# Epidemiological study on foot-and-mouth disease in small ruminants: Sero-prevalence and risk factor assessment in Kenya

**Eunice C. Chepkwony**[1], **George C. Gitao**[2], **Gerald M. Muchemi**[3], **Abraham K. Sangula**[1], **Salome W. Kairu-Wanyoike**[4]*

**1** Foot and Mouth Disease National Laboratory, Embakasi, Directorate of Veterinary Services, State Department of Livestock, Nairobi, Kenya, **2** Department of Veterinary Pathology, Microbiology and Parasitology, University of Nairobi, Nairobi, Kenya, **3** Department of Public Health, Pharmacology and Toxicology, University of Nairobi, Nairobi, Kenya, **4** Meat Training Institute, Directorate of Veterinary Services, State Department of Livestock, Nairobi, Kenya

* swwanyoike@yahoo.com

**Data Availability Statement:** All relevant data are within the manuscript and its S1 File, S1–S3 Tables files.

## Abstract

Foot-and-mouth disease (FMD) is endemic in Kenya affecting cloven-hoofed ruminants. The epidemiology of the disease in small ruminants (SR) in Kenya is not documented. We carried out a cross-sectional study, the first in Kenya, to estimate the sero-prevalence of FMD in SR and the associated risk factors nationally. Selection of animals to be sampled used a multistage cluster sampling approach. Serum samples totaling 7564 were screened for FMD antibodies of non-structural-proteins using ID Screen® NSP Competition ELISA kit. To identify the risk factors, generalized linear mixed effects (GLMM) logistic regression analysis with county and villages as random effect variables was used. The country animal level sero-prevalence was 22.5% (95% CI: 22.3%-24.3%) while herd level sero-prevalence was 77.6% (95% CI: 73.9%-80.9%). The risk factor that was significantly positively associated with FMD sero-positivity in SR was multipurpose production type (OR = 1.307; p = 0.042). The risk factors that were significantly negatively associated with FMD sero-positivity were male sex (OR = 0.796; p = 0.007), young age (OR = 0.470; p = 0.010), and sedentary production zone (OR = 0.324; p<0.001). There were no statistically significant intra class correlations among the random effect variables but interactions between age and sex variables among the studied animals were statistically significant (p = 0.019). This study showed that there may be widespread undetected virus circulation in SR indicated by the near ubiquitous spatial distribution of significant FMD sero-positivity in the country. Strengthening of risk-based FMD surveillance in small ruminants is recommended. Adjustment of husbandry practices to control FMD in SR and in-contact species is suggested. Cross-transmission of FMD and more risk factors need to be researched.

**Funding:** This study had support from a project titled "Improving Animal Disease Surveillance in Support of Trade in IGAD Member States", in short "Surveillance of Trade Sensitive Diseases – STSD". This was a regional component of the Supporting the Horn of Africa's Resilience (SHARE). The project was implemented by IGAD member states through AU-IBAR and IGAD. The project was implemented with financial support from the European Union (EU). The direct recipient of the funding in Kenya was the Directorate of Veterinary Services with SWK as the National Focal Person. The ELISA Kits used in this work on FMD in small ruminants were provided by Eu-FMD through the Nakuru FMD Real-Time Training Course credit points. The direct recipient of the kits was ECC at the FMD laboratory of the Directorate of Veterinary Services. The AU-IBAR and IGAD Secretariat played a role in study design as well as training on data collection and data management in the main project. Sample testing, data analysis and publication was not funded by the main project as it fell outside the scope of the project.

**Competing interests:** The authors have declared that no competing interests exist.

## Introduction

Livestock husbandry in developing countries like Kenya is critical for ensuring food security and for poverty alleviation [1]. Livestock are a source of meat, milk, hides and compost manure as well as an insurance against emergencies [1–3]. Sheep and goats (small ruminants) are sometimes preferred by farmers compared to large ruminants because of the small space they occupy and less fodder requirement. In addition, goats have high adaptability to harsh climates which makes them suitable for husbandry in marginal areas [2, 3]. Small ruminant population in Kenya stands at 17.1 million sheep, 27.7 million goats about 50–57% of which are in the pastoral and agro-pastoral production areas [4, 5]. Sheep breeds include red maasai, black-head Persian and east African fat tailed sheep. Among goats, the small east African is most dominant although milk breeds such as the Galla and Toggenberg are also to be found [6].

Infectious diseases constrain small ruminant (SR) production [7]. The S1 Table shows the sero-prevalence to FMD in small ruminants and associated risk factors that have been reported in various countries [8–21]. Therefore, in East African countries, FMD sero-prevalence in SR of between 4.0% and 48.5% has been reported. The risk factors that have been associated with FMD sero-positivity in SR in these countries are agro-ecology, production system, age, sex, contact with wildlife, season, breed, interaction with other livestock species, herd size, acquisition of livestock and husbandry practices [8, 9, 15, 18, 19, 20, 21].

Foot and mouth disease (FMD) is an acute highly contagious, transboundary, disease caused by foot and mouth disease virus (FMDV). It affects cloven-hoofed domestic ruminants and pigs, as well as wild ruminants [22]. It severely affects livestock production leading to disruption of trade in animals and their products at regional and international level. A global strategy for the control of FMD was endorsed in 2012 to minimize the burden of FMD in endemic settings and maintain free status in FMD-free countries [23].

The FMDV is classified into the *Picornaviridae* family and the genus *Apthovirus*. It is a small non-enveloped virus with an encoder for four structural proteins and ten nonstructural proteins [24]. The disease is among the World Organisation for Animal Health (OIE) listed diseases requiring immediate reporting and investigation in order to control its spread [23].

The incubation period for foot-and-mouth disease is 3–8 days in small ruminants [25]. The disease is characterized by high fever within two to three days, formation of vesicles and erosions inside the mouth leading to drooling of saliva. Vesicles are also on the nose, teats and when on the feet may rupture and cause lameness. It also causes several months of weight loss in adults and significant temporary or permanent reduction in milk production [26]. In sheep the disease persists for up to nine months and in goats for up to six months [27]. Foot and mouth disease in adult sheep and goats is frequently asymptomatic, but can cause high mortality in young animals. Clinical disease in young lambs and kids is characterized by death without the appearance of vesicles, due to heart failure following myocarditis [28]. Lameness is often characterized by unwillingness to rise and move [25, 29]. The disease can easily be missed unless individual animals are carefully examined for disease lesions. Small ruminants can therefore be responsible for the introduction of FMD into previously disease-free herds [30].

Although FMD may be suspected based on clinical signs and post-mortem findings, it cannot be differentiated clinically from other vesicular diseases [26]. Confirmation of any suspected FMD case through laboratory tests is therefore essential. Detection of the antibodies against the non-structural proteins (NSPs) of FMDV is used for differentiation between infected and vaccinated animals (DIVA) which is of great importance in the FMD control program. The 3ABC competition antibody ELISA which has high sensitivity and specificity can

deliver same-day results when using the short protocol and is routinely applied for general screening for FMD [31, 32].

Foot and mouth disease is endemic in Kenya with outbreaks in cattle and serotype O has been the most prevalent serotype. Intermittent circulation of FMDV serotypes A, SAT 1 and SAT 2 have also been confirmed in various parts of the country in the last five years [33]. The main FMD control strategies in the country focus on vaccination of cattle. Although small ruminants are also affected by FMD and are herded together with cattle, they are not usually vaccinated [34].

Some studies have been carried out on FMD in cattle and buffaloes but no studies on the prevalence and associated risk factors in small ruminants have been done in Kenya. This study investigated the sero-prevalence and potential risk factors associated with FMD in domestic small ruminants in Kenya.

## Materials and methods

### The study population

The study was a cross-sectional one which targeted the national small ruminant population in Kenya. Kenya is made up of 47 counties. However, the objective was not to primarily measure the sero-prevalence per county but rather per the major small ruminant production zones. The sampling unit was the smallest administrative unit in record, the village, which was selected after first selecting the second smallest administrative unit, the sub-location. The sampling frame of sub-locations was available from the Kenya 2009 population and housing census [5].

### Description of the study area

Broadly, Kenya can be divided into three ecological zones namely humid, semi-arid and arid areas. About 80% of the country is arid and semi-arid (ASAL) while the humid ecosystem occupies the remaining 20% of the country. The semi-arid areas normally experience short rainfall with prolonged drought while arid areas have long cyclic droughts, thus affecting pasture and water availability. The humid areas have long rain seasons with heavy down pours reaching 2700mm [35].

The main small ruminant production systems are pastoralism and agro-pastoralism as well as sedentary/mixed systems. Pastoral systems are in the arid and semi-arid climate zones where about 14 million people are dependent on livestock [36, 37]. Agro-pastoralism, is livestock production which is associated with dryland or rain-fed cropping and animals range over short distances. The average herd size of sheep and goat in pastoralist systems is estimated at 24.9 and 75.2 respectively [38]. Sedentary/mixed systems are found in the semi-arid, sub-humid, humid and highland zones. This farming system is based on livestock but practiced in proximity to, or perhaps in functional association with, other farming systems based on cropping or is a livestock subsystem of integrated crop-livestock farming. The average herd size of sheep and goats in this system is rarely reported but ranges between three and 10 [4, 37, 39]

### Study design and methods

**Sample size determination.** For the purpose of this study, the country was divided into two zones; the pastoral zone (PZ) and the sedentary zone (SZ) as in Fig 1. Sample size calculation was in two stages: number of herds to be sampled and then number of animals to be sampled per herd. A herd included a group of sheep and goats in a farm and animals from neighbouring farms which came into contact (in-contact farms). The formula used was that by

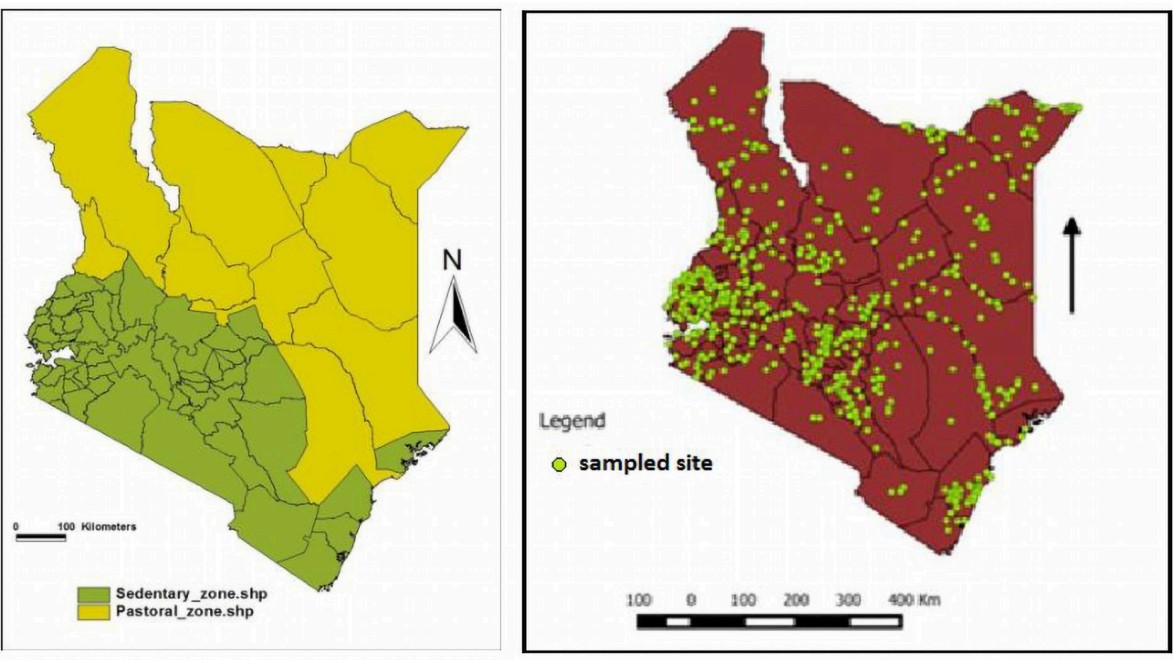

**Fig 1. Study zones and selected sampling sites for the cross-sectional survey, Kenya, 2016.**

Humphry et al [40]. The assumptions that were made were: number of herds >10,000 in each zone; confidence level = 95%; accepted margin of error = 5%; expected proportion of positive herds in the population = 70% (expert opinion, confirmed now by Ahmed et al [41]; intra-class correlation coefficient (measure of variation between clusters) = 0.1 [42]; design effect = 2 [43]; test specificity = 99% [23]; test sensitivity = 100% [23]. The calculated sample size was 323 x design effect (2) = 646 herds countrywide. The minimum number of animals to be sampled per herd was determined as summarized in Cannon and Roe table cited by Thrusfield (p.239) [44]. This was determined making the following assumptions: expected prevalence of FMD in the herd = 20%; which was an average of two consecutive years in Tanzania [19]; confidence level = 95%; average herd size = 100. This yielded a sample size of 13 animals per herd which was increased to 14 to take care of any possible losses. Countrywide therefore, 646 herds resulting from one village per sub-location and one herd per village (or more if necessary to obtain sufficient animals) and 14 animals per herd yielded a sample size of 646 x 14 = 9044 samples. The two zones are quite distinct in structure even if FMD dynamics (in cattle) seem not very different. The SZ has small counties with very many sub-locations while the PZ has large counties with fewer sub-locations which could have led to over sampling in the SZ and low sampling in the PZ. It was also important to see the difference in the two zones and for this reason we chose to have a complete separation between the two and sample equal number of sub-locations (323) and therefore number of samples (4522) in each zone.

**Sampling of herds and animals.**   The sampling frame consisted of the 6796 sub-locations, as obtained from the national census of 2009 [5]. The sampling frame was used to randomly select the sub-locations to be sampled. However, sub-locations can be quite large and therefore once in the field the teams obtained the list of villages in the sub-locations and randomly selected villages within the sub-locations. A herd was then considered as all animals within the village from which the individual animals sampled were randomly selected. If one herd could yield all the animals required, only that herd was sampled, otherwise additional herds close to

the selected herd were sampled until the required number to be sampled in the sub-location was reached. Therefore, a multistage cluster sampling was employed to select the animals from the two zones.

**Data and serum sample collection.** Data were collected using questionnaires and as laboratory results while serum samples were collected from eligible herds throughout the country.

*Serum sample collection.* Serum samples were collected by 15 teams of trained laboratory technicians under the supervision of a veterinarian. Sheep and goats aged six months and above were sampled to avoid those with maternal antibodies [45]. Age was determined by examining the dentition of each animal and information from the owner for young animals with no permanent incisors [46]. In the sampling stage animal level variables (biodata) were collected into a sampling form (S1 File) and included species (ovine or caprine), breed, age and the sex of the animal and origin (whether born in herd or brought in). The blood samples were collected from a jugular vein, using 10 ml sterile vacutainer tubes and gauge 21 needles and labeled with a unique identification (county code/sub-location/animal number/sex/age). The samples were then allowed to clot in cool-boxes. Once the blood clot had retracted after 12 to 24 hours the vials were centrifuged in the laboratories in the field to obtain serum which was placed in two 2ml cryovials (two aliquots) labeled with corresponding identification codes. In areas where laboratories or centrifuges were unavailable, serum was separated using sterile disposable pipettes (one per sample) and transferred into the cryovials. Samples were stored at -20˚C in freezers located in the areas sampled in the field until the end of the sampling period (which was not more than 20 days per team) and were transported on ice using cool-boxes to the Central Veterinary Laboratories (CVL), Kabete, Kenya. At the CVL, the samples were held at -86˚C until testing after which they were placed in a serum bank at the same temperature. Sample collection was part of a larger national survey for Rift Valley Fever (RVF) and Peste de Petits Ruminants (PPR) antibodies in small ruminants under a project titled "Improving Animal Disease Surveillance in Support of Trade in IGAD Member States", in short "Surveillance of Trade Sensitive Diseases–STSD". One aliquot was used to test for the presence or absence of antibodies for Rift Valley Fever (RVF) and Peste de Petits Ruminants (PPR) antibodies according to the objective of the STSD project. The second aliquot was moved to the FMD National Laboratory, Embakasi, Kenya and stored at -20˚C until laboratory investigation for FMD antibodies.

*Questionnaire administration.* A pre-tested semi-structured questionnaire (S1 File) was administered in-person by trained enumerators to owners of sampled herds following the guidelines (S1 File) at the time of sample collection for collection of herd-level variables. Herd-level variables were production zone, whether the herd owners brought in animals in the last one year, whether the herd owners purchased animals from the market/middlemen, interaction with wildlife, production type, production system, housing type, grazing system, watering system, breeding method and altitude/elevation. Also based on Geographic Positioning System (GPS) technology, GPS coordinates and elevations were recorded for each herd location and this information was recorded in each questionnaire form which was labeled with the unique herd identification code.

**Laboratory sample analysis.** Individual animal serum samples were analysed using the foot and mouth disease virus 3 ABC- ELISA ID Screen® FMD NSP Competition kit (ID-VET, Grabels, France) to detect specific antibodies against the non-structural protein (NSP) of FMDV regardless of sero-type. This was done according to the manufacturer's protocol. The test has specificity of 99% and sensitivity of 100% [23]. A herd was considered as positive if one or more animals in the herd were seropositive.

**Data management and analysis.** Individual animal laboratory data generated during testing along with individual animal biodata data obtained during sample collection (species,

breed, sex, age, origin) were entered in Microsoft Excel 2010 spreadsheet. Questionnaire data which included mainly herd data (county type, production zone, whether animals were brought into the herd, whether animals were purchased from markets and middlemen, wildlife interaction, production type, production system, housing system, grazing system, watering system, breeding method and altitude/elevation) were entered in Microsoft Access 2010 due to the large amount of data and need to link the data tables. The required data columns from each data set were then brought together in a Microsoft Excel Spreadsheet, data cleaned and coded before being exported for descriptive analysis using IBM Statistical Package for Social Science (SPSS) Statistics for Windows Version 20 (IBM Corp., Armonk, N.Y., USA) and R version 4.0.3 (2020-10-10) for regression analysis. Descriptive analysis generated sums, means, proportions and confidence intervals. Descriptive statistics were also generated for the sero-prevalence in the two different production zones (pastoral and sedentary), for each county and for the other potential risk factors. Apparent prevalence was calculated using Eq 1 [44] while true prevalence was calculated using Eq 2 [47]. Confidence interval of the true prevalence was calculated using Eq 3 [48, 49].

$$Apparent\ prevalence\ =\ \frac{No.\ of\ animals\ testing\ positive}{Total\ number\ of\ animals\ in\ the\ group\ tested} x\ 100 \tag{1}$$

$$True\ prevalence\ =\ \frac{apparent\ prevalence + specificity - 1}{sensitivity + specificity - 1} x 100 \tag{2}$$

$$(95\%CI\ of\ true\ prevalence\ =\ p \pm 1.96\sqrt{(\tfrac{pq}{nJ^2})}. \tag{3}$$

Where, p is apparent prevalence; q is 1-p; n is sample size and $J^2$ is Youden's index (Se+Sp-1) where Se is test sensitivity and Sp is test specificity.

Chi-squared test as recommended by Campbell [50] and Richardson [51] was used for pairwise comparison of proportions while the confidence intervals of the proportions were calculated using the method recommended by Altman et al. [52]. Coding in regression analyses was such that the lowest code (0), the reference, was the factor which exhibited the highest proportion/Wald statistic [53]. Test for collinearity of the variables was by testing for correlation. Simple correlation coefficients for pairs of independent variables are determined and a value of more than 0.3 was considered reasonable collinearity among a pair of independent variables and one was dropped [54]. This was done systematically until only those with correlation of 0.3 or less remained. Multivariable generalized linear mixed effects logistic regression analysis (GLMM) with county and villages as random effect variables was used to test the strength of association between the potential risk factors and FMD sero-positivity. This made use of backward fitting of variables and generated odds ratio (OR) and p values. Interaction between variables was also tested. The interpretation of odds ratios less than one were after obtaining their inverse [55]. Scaled residuals and fitted values were generated and used to evaluate the final models developed. In all the analysis, confidence level was kept at 95% and ρ≤0.05 was set for significance.

The goodness of fit test used for the regression models was the Akaike Information Criterion (AIC) which maximizes the likelihood function. The model with the lowest AIC was considered as the most parsimonious [56, 57].

**Ethics.** The research approval for the study was obtained from the Kenya National Commission for Science, Technology and Innovation (NACOSTI/P/19/57224/31389) and the Faculty of Veterinary Medicine Biosafety, Animal Use and Ethics Committee (FVM BAUEC/ 2020/262). Each owner of a herd selected for sampling provided verbal consent, once the

objectives of the study were explained. Herds whose owners did not consent were replaced with the next herd in the random sample list. Other approvals required for the study were obtained from the State Department of Livestock at national level and from the respective county governments.

## Results

The cross-sectional study was carried out from August to September 2016 cross-nationally. In the study, 898 herds were sampled yielding 8201 samples. The herds were more than the number calculated since, especially in the sedentary area, it was difficult to find sufficient animals in one herd. However, only 7564 samples from 872 herds were available for testing for FMD sero-prevalence as 637 were already depleted while testing for other diseases or had spilled or were grossly contaminated. Sheep samples were 2560 (33.8%) while goat samples were 5004 (66.2%). Of these 3909 (51.7%) were from the PZ and 3655 (48.3%) were from the SZ. Of the 44 counties investigated (samples from three out of 47 counties not available), 11 (25.0%) were in the PZ and 33 (75.0%) were in the SZ.

### Animal and herd level descriptive statistics

S2 Table shows the number of sheep and goats in the sampled herds in the PZ and SZ. Therefore, the mean herd size in the PZ was about ten times that in the SZ. The animals in both zones were mainly females older than one year.

Table 1 presents the individual animal variable descriptive statistics in both the PZ and SZ and overall. Thus about two-thirds of the SR sampled was of caprine species. Nearly half of the animals were of local breed. About three-quarters of the SR sampled were female. Majority of the animals sampled were mature (97.0%) and born in the herds (82.9%).

Table 2 presents the herd level variable descriptive statistics in both zones and overall. Although these were herd level variables, the numbers are of actual number of animals involved as most analyses in the study were at individual animal level. Overall about two-thirds of animals were in herds where SR were brought into the herds in the last one year, in herds which purchased SR from markets or middlemen, in herds under communal grazing, in herds which shared watering and in herds at altitude of less than 1500m above sea level. Just over a half of the animals were from herds which had interaction with wildlife and from herds which utilized own-male breeding method. Majority of animals were in meat and multi-purpose production type (86.1%), in the sedentary and pastoral production system (75.5%) and were enclosed at night (77.0%).

### Sero-prevalence of FMD in small ruminants

At the time of sampling, none of the animals in the surveyed herds showed FMD clinical symptoms. Sero-prevalence of non-structural FMDV protein (antibodies) for a total of 7564 sera collected from the whole country was determined. The overall true sero-prevalence of FMD in small ruminants was 22.5% (95% CI: 22.3–24.3%). The sero-prevalence was significantly higher (p = 0.021) in the PZ at 31.2% (95% CI: 29.8–32.7%) compared to that in the SZ which had a sero-prevalence of 14.7% (95% CI: 13.4–15.7%). The sero-prevalence per county is in the S3 Table. The distribution of FMD sero-positives among SR was near ubiquitous with nearly every county registering some positives. Variations in spatial distributions of FMD sero-prevalence were observed across the country with true sero-prevalence levels higher than the national average of 22.5% recorded in 11 (25%) of counties among them Mandera, Kilifi, Lamu, Kajiado, West-Pokot, Garissa, Turkana, Wajir, Kwale Tana River and Isiolo counties, mainly in the PZ except for Kilifi, Lamu and Kwale counties. Some counties (14), mainly in

**Table 1. Descriptive statistics of sampled individual animal variables in the pastoral and sedentary zones, Kenya, 2016.**

| Variable | No. in Pastoral Zone | % in Pastoral Zone | No. in Sedentary Zone | % in Sedentary Zone | Total | % of Total |
|---|---|---|---|---|---|---|
| **Species** | | | | | | |
| Caprine | 2694 | 68.9 | 2310 | 63.2 | 5004 | 66.2 |
| Ovine | 1215 | 31.1 | 1345 | 36.8 | 2560 | 33.8 |
| *Total* | 3909 | 100.0 | 3655 | 100.0 | 7564 | 100.0 |
| **Breed** | | | | | | |
| Local | 1714 | 43.8 | 1596 | 43.7 | 3310 | 43.8 |
| Cross-breed | 431 | 11.0 | 430 | 11.8 | 861 | 11.4 |
| Exotic | 178 | 4.6 | 516 | 14.1 | 694 | 9.2 |
| Unidentified | 1586 | 40.6 | 1113 | 30.5 | 2699 | 35.7 |
| *Total* | 3909 | 100.0 | 3655 | 100.0 | 7564 | 100.0 |
| **Sex** | | | | | | |
| Female | 3010 | 77.0 | 2912 | 79.7 | 5922 | 78.3 |
| Male | 894 | 22.9 | 742 | 20.3 | 1636 | 21.6 |
| Unidentified | 5 | 0.1 | 1 | 0.0 | 6 | 0.1 |
| *Total* | 3909 | 100.0 | 3655 | 100.0 | 7564 | 100.0 |
| **Age** | | | | | | |
| Mature (>1 year) | 3879 | 99.2 | 3456 | 94.6 | 7335 | 97.0 |
| Young (≤1 year) | 11 | 0.3 | 187 | 5.1 | 198 | 2.6 |
| Unidentified | 19 | 0.5 | 12 | 0.3 | 31 | 0.4 |
| *Total* | 3909 | 100.0 | 3655 | 100.0 | 7564 | 100.0 |
| **Origin** | | | | | | |
| Born in herd | 3145 | 80.5 | 3128 | 85.6 | 6273 | 82.9 |
| Brought in | 50 | 1.3 | 267 | 7.3 | 317 | 4.2 |
| Unidentified | 714 | 18.3 | 260 | 7.1 | 974 | 12.9 |
| *Total* | 3909 | 100.0 | 3655 | 100.0 | 7564 | 100.0 |

the SZ namely Embu, Kisii, Nakuru, Elgeiyo- Marakwet, Kiambu, Bungoma, Kirinyaga, Vihiga and Murang'a counties had sero-prevalence of less than 10.0%. Other counties (19) in sedentary zone had sero-prevalence above 10.0% but lower than the national average. The sero-prevalence for Mombasa and Nyamira was 0.0% but the number of samples tested was too small (5 and 14 respectively) to give any meaningful interpretation.

The FMD sero-positivity per potential individual animal risk factor was as in Table 3.

For variables with more than two categories, chi-square reported in Tables 3 and 4 is for sero-prevalence for all categories. The chi-square reported in the ensuing text are for pair-wise comparison of sero-prevalence. Thus at individual animal level, the sero-positivity of FMD in caprine (goats) compared to that in ovine (sheep) was significantly higher (p<0.001). That for exotic breeds was significantly lower than that for local breeds ($\chi2 = 14.43$; p<0.001) and cross breeds ($\chi2 = 9.13$; p = 0.003). Sero-prevalence in mature animals was significantly higher than in young animals (p<0.001) while that in animals that were born in the herd was significantly higher than that of animals that were brought in (p<0.001).

The herd level prevalence, a measure of sero-prevalence in herds where at least one animal in a herd tested positive, for all the 872 herds tested was 77.6% (95% CI: 73.9–80.9), which was significantly higher than overall animal level sero-prevalence (22.5% (95% CI: 22.3–24.3). The sero-positivity per potential herd risk factor was as in Table 4.

Herds which had brought in SR in the last one year had significantly lower sero-prevalence than those that had not (p<0.001). Herds in which animals were bought from the market or

**Table 2. Descriptive statistics of herd level variables in the pastoral and sedentary zones, Kenya, 2016.**

| Variable | No. in PZ | % in PZ | No. in SZ | % in SZ | Total | % of Total |
|---|---|---|---|---|---|---|
| **Herds brought in SR?** | | | | | | |
| No | 2494 | 63.8 | 2273 | 62.2 | 4767 | 63.0 |
| Yes | 1387 | 35.5 | 1382 | 37.8 | 2769 | 36.6 |
| Unidentified | 28 | 0.7 | 0 | 0.0 | 28 | 0.4 |
| *Total* | 3909 | 100.0 | 3655 | 100.0 | 7564 | 100.0 |
| **Buy SR from market or middlemen?** | | | | | | |
| No | 1733 | 44.3 | 3020 | 82.6 | 4753 | 62.8 |
| Yes | 2176 | 55.7 | 635 | 17.4 | 2811 | 37.2 |
| *Total* | 3909 | 100.0 | 3655 | 100.0 | 7564 | 100.0 |
| **SR interaction with wildlife** | | | | | | |
| Yes | 3246 | 83.0 | 1053 | 28.8 | 4299 | 56.8 |
| No | 209 | 5.3 | 1242 | 34.0 | 1451 | 19.2 |
| Unidentified | 454 | 11.6 | 1360 | 37.2 | 1814 | 24.0 |
| *Total* | 3909 | 100.0 | 3655 | 100.0 | 7564 | 100.0 |
| **SR Production type** | | | | | | |
| Meat | 1594 | 40.8 | 1747 | 47.8 | 3341 | 44.2 |
| Multipurpose | 1876 | 48.0 | 1294 | 35.4 | 3170 | 41.9 |
| Mixed | 228 | 5.8 | 138 | 3.8 | 366 | 4.8 |
| Dairy | 20 | 0.5 | 170 | 4.7 | 190 | 2.5 |
| Unidentified | 191 | 4.9 | 306 | 8.4 | 497 | 6.6 |
| *Total* | 3909 | 100.0 | 3655 | 100.0 | 7564 | 100.0 |
| **SR Production System** | | | | | | |
| Sedentary/mixed | 265 | 6.8 | 2850 | 78.0 | 3115 | 41.2 |
| Pastoral | 2481 | 63.5 | 112 | 3.1 | 2593 | 34.3 |
| Agro-pastoral | 782 | 20.0 | 266 | 7.3 | 1048 | 13.9 |
| Multiple | 138 | 3.5 | 28 | 0.8 | 166 | 2.2 |
| Unidentified | 243 | 6.2 | 399 | 10.9 | 642 | 8.5 |
| *Total* | 3909 | 100.0 | 3655 | 100.0 | 7564 | 100.0 |
| **SR Housing** | | | | | | |
| Enclosed at night | 3061 | 78.3 | 2760 | 75.5 | 5821 | 77.0 |
| None | 794 | 20.3 | 592 | 16.2 | 1386 | 18.3 |
| Enclosed day and night | 54 | 1.4 | 303 | 8.3 | 357 | 4.7 |
| *Total* | 3909 | 100.0 | 3655 | 100.0 | 7564 | 100.0 |
| **SR grazing** | | | | | | |
| Communal | 1611 | 41.2 | 1113 | 30.5 | 2724 | 36.0 |
| Fenced | 318 | 8.1 | 1799 | 49.2 | 2117 | 28.0 |
| Mixed | 1070 | 27.4 | 198 | 5.4 | 1268 | 16.8 |
| Migratory | 744 | 19.0 | 42 | 1.1 | 786 | 10.4 |
| Unidentified | 124 | 3.2 | 234 | 6.4 | 358 | 4.7 |
| Zero-grazing | 42 | 1.1 | 269 | 7.4 | 311 | 4.1 |
| *Total* | *3909* | 100.0 | *3655* | 100.0 | 7564 | 100.0 |
| **SR watering** | | | | | | |
| Shared | 3529 | 90.3 | 1011 | 27.7 | 4540 | 60.0 |
| On-farm | 199 | 5.1 | 2226 | 60.9 | 2425 | 32.1 |
| Unidentified | 181 | 4.6 | 274 | 7.5 | 455 | 6.0 |
| Mixed | 0 | 0.0 | 144 | 3.9 | 144 | 1.9 |
| *Total* | 3909 | 100.0 | 3655 | 100.0 | 7564 | 100.0 |

*(Continued)*

**Table 2.** (Continued)

| Variable | No. in PZ | % in PZ | No. in SZ | % in SZ | Total | % of Total |
|---|---|---|---|---|---|---|
| **SR breeding method** | | | | | | |
| Own male | 2059 | 52.7 | 2370 | 64.8 | 4429 | 58.6 |
| Mixed | 1014 | 25.9 | 230 | 6.3 | 1244 | 16.4 |
| Common-use male | 556 | 14.2 | 244 | 6.7 | 800 | 10.6 |
| Unidentified | 183 | 4.7 | 395 | 10.8 | 578 | 7.6 |
| Male from another farm | 64 | 1.6 | 382 | 10.5 | 446 | 5.9 |
| Artificial insemination | 33 | 0.8 | 34 | 0.9 | 67 | 0.9 |
| *Total* | 3909 | 100.0 | 3655 | 100.0 | 7564 | 100.0 |
| **SR location elevation** | | | | | | |
| ≤1500m | 3130 | 80.1 | 1754 | 48.0 | 4884 | 64.6 |
| >1500m | 641 | 16.4 | 1828 | 50.0 | 2469 | 32.6 |
| Unidentified | 138 | 3.5 | 73 | 2.0 | 211 | 2.8 |
| *Total* | 3909 | 100.0 | 3655 | 100.0 | 7564 | 100.0 |

middlemen had significantly higher sero-positivity than in those herds where this was not the case (p<0.001). Similarly, herds which had wildlife interaction had significantly higher sero-positivity than those without such interaction (p<0.001). The sero-positivity of herds at low altitude (≤1500m above sea level was significantly higher (p<0.001) than that of herds at higher altitude (>1500m above sea level).

Multipurpose production type herds had significantly higher sero-positivity than meat ($\chi^2$ = 20.00; p<0.001), mixed ($\chi^2$ = 7.62; p = 0.006) and dairy ($\chi^2$ = 8.41; p = 0.004) production

**Table 3. FMD Sero-positivity per potential individual animal risk factor, Kenya, 2016.**

| Variable | Total tested | Positive | % Positive | 95%CI of % positive | $\chi 2$ | p-value |
|---|---|---|---|---|---|---|
| **Species** | | | | | | |
| Caprine | 5004 | 1202 | 24.0 | 22.9–25.2 | 789.68 | <0.001 |
| Ovine | 2560 | 560 | 21.9 | 20.3–23.5 | | |
| **Breed** | | | | | | |
| Local | 3310 | 807 | 24.4 | 22.9–25.9 | 2728.79 | <0.001 |
| Cross-breed | 861 | 207 | 24.0 | 21.2–27.1 | | |
| Exotic | 694 | 123 | 17.7 | 15.0–20.8 | | |
| Unidentified[φ] | 2699 | 25 | 0.9 | 0.6–1.4 | | |
| **Sex** | | | | | | |
| Female | 5922 | 1403 | 23.7 | 22.6–24.8 | 7406.90 | <0.001 |
| Male | 1636 | 356 | 21.8 | 19.8–23.9 | | |
| Unidentified | 6 | 3 | 50.0 | 14.0–86.1 | | |
| **Age** | | | | | | |
| Mature (>1 year) | 7335 | 1731 | 23.6 | 22.6–24.6 | 13790.73 | <0.001 |
| Young (≤1 year) | 198 | 20 | 10.1 | 6.4–15.4 | | |
| Unidentified | 31 | 20 | 35.5 | 19.8–54.6 | | |
| **Origin** | | | | | | |
| Born in herd | 6273 | 1474 | 23.5 | 22.5–24.6 | 8459.147 | <0.001 |
| Brought in | 317 | 54 | 17.0 | 13.2–21.7 | | |
| Unidentified | 974 | 234 | 24.0 | 21.4–26.9 | | |

[φ]Unidentified means that the variable was not indicated for those samples. The chi-square and p value are overall values for the differences in proportions.

**Table 4. Sero-positivity for FMD per potential herd risk factor, Kenya, 2016.**

| Variable | Total tested | Positive | % Positive | 95%CI of % positive | $\chi^2$ | p-value |
|---|---|---|---|---|---|---|
| **Production Zone** | | | | | | |
| Pastoral (PZ) | 3909 | 1220 | 31.2 | 29.8–32.7 | 5.29 | 0.021 |
| Sedentary (SZ) | 3655 | 531 | 14.7 | 13.4–15.7 | | |
| **Herds brought in SR?** | | | | | | |
| No | 4767 | 1160 | 24.3 | 23.1–25.6 | 4490.11 | <0.001 |
| Yes | 2769 | 598 | 21.6 | 20.1–23.2 | | |
| Unidentified[φ] | 28 | 4 | 14.3 | 4.7–33.6 | | |
| **Buy SR from market or middlemen?** | | | | | | |
| No | 4753 | 1023 | 21.5 | 20.4–22.7 | 498.59 | <0.001 |
| Yes | 2811 | 739 | 26.3 | 24.7–28.0 | | |
| **SR interaction with wildlife** | | | | | | |
| Yes | 4299 | 1201 | 27.9 | 26.6–29.3 | 1906.15 | <0.001 |
| No | 1451 | 183 | 12.6 | 11.0–14.5 | | |
| Unidentified | 1814 | 378 | 20.8 | 19.0–22.8 | | |
| **SR Production type** | | | | | | |
| Meat | 3341 | 700 | 21.0 | 19.6–22.4 | 6732.83 | <0.001 |
| Multipurpose | 3170 | 816 | 25.7 | 24.2–27.3 | | |
| Mixed | 366 | 70 | 19.1 | 15.3–23.6 | | |
| Dairy | 190 | 31 | 16.3 | 11.5–22.5 | | |
| Unidentified | 497 | 145 | 29.2 | 25.3–33.4 | | |
| **SR Production System** | | | | | | |
| Sedentary/mixed | 3115 | 463 | 14.9 | 13.6–16.2 | 4311.26 | <0.001 |
| Pastoral | 2593 | 799 | 30.8 | 29.0–32.6 | | |
| Agro-pastoral | 1048 | 305 | 29.1 | 26.4–32.0 | | |
| Mixed | 166 | 37 | 22.3 | 16.4–29.5 | | |
| Unidentified | 642 | 158 | 24.6 | 21.4–28.2 | | |
| **SR Housing** | | | | | | |
| Enclosed at night | 5821 | 1377 | 23.7 | 22.6–24.8 | 6687.38 | <0.001 |
| None | 1386 | 346 | 25.0 | 22.7–27.3 | | |
| Enclosed day and night | 357 | 39 | 10.9 | 8.0–14.7 | | |
| **SR grazing** | | | | | | |
| Communal | 2724 | 713 | 26.2 | 24.5–27.9 | 3820.75 | <0.001 |
| Fenced | 2117 | 335 | 15.8 | 14.3–17.5 | | |
| Mixed | 1268 | 351 | 27.7 | 25.3–30.3 | | |
| Migratory | 786 | 232 | 29.5 | 26.4–32.9 | | |
| Unidentified | 358 | 96 | 26.8 | 22.4–31.8 | | |
| Zero-grazing | 311 | 35 | 11.3 | 8.1–15.4 | | |
| **SR watering** | | | | | | |
| Shared | 4540 | 1290 | 28.4 | 27.1–29.8 | 6566.08 | <0.001 |
| On-farm | 2425 | 330 | 13.6 | 12.3–15.1 | | |
| Unidentified | 455 | 124 | 27.3 | 23.3–31.6 | | |
| Mixed | 144 | 18 | 12.5 | 7.8–19.3 | | |
| **SR breeding method** | | | | | | |
| Own male | 4429 | 1020 | 23.0 | 21.8–24.3 | 10157.63 | <0.001 |
| Mixed | 1244 | 325 | 26.1 | 23.7–28.7 | | |
| Common-use male | 800 | 215 | 26.9 | 23.9–30.1 | | |
| Unidentified | 578 | 131 | 22.7 | 19.4–26.3 | | |

(*Continued*)

**Table 4.** (Continued)

| Variable | Total tested | Positive | % Positive | 95%CI of % positive | $\chi^2$ | p-value |
|---|---|---|---|---|---|---|
| Male from another farm | 446 | 63 | 14.1 | 11.1–17.8 | | |
| Artificial insemination | 67 | 8 | 11.9 | 5.7–22.7 | | |
| **SR location elevation** | | | | | | |
| ≤1500m | 4884 | 1277 | 26.2 | 24.9–27.4 | 4332.06 | <0.001 |
| >1500m | 2469 | 419 | 17.0 | 15.5–18.5 | | |
| Unidentified | 211 | 66 | 31.3 | 25.2–38.1 | | |

<sup>φ</sup>Unidentified means that the variable was not indicated for those samples. The chi-square and p value are overall values for the differences in proportions.

type herds. The pastoral production system showed significantly higher sero-positivity than sedentary ($\chi^2$ = 207.61; p<0.001), agro-pastoral ($\chi^2$ = 104.96;p<0.001) and mixed ($\chi^2$ = 5.34; p = 0.021) production systems. The sero-prevalence in the sedentary production system was significantly higher than that in the mixed production system ($\chi^2$ = 6.67; p = 0.001). Herds which were not enclosed or enclosed only at night had a significantly higher sero-positivity than herds which were enclosed by day and by night ($\chi^2$ = 32.75; p<0.001 and $\chi^2$ = 31.15; p<0.001 respectively). Communal grazed herds had significantly higher sero-prevalence than fenced ($\chi^2$ = 75.94; p<0.001) and zero-grazed herds ($\chi^2$ = 33.33; p<0.001). Fenced herds had significantly higher sero-prevalence than herds with mixed grazing ($\chi^2$ = 69.50; p<0.001), herds with migratory grazing ($\chi^2$ = 68.49) and zero-grazed herds ($\chi^2$ = 4.25; p = 0.04). Herds with mixed and migratory grazing systems had significantly higher sero-prevalence than zero-grazed herds ($\chi^2$ = 36.32; p<0.001 and $\chi^2$ = 40.04; p<0.001 respectively) Small ruminant herds that had shared watering had significantly higher sero-positivity than those with on farm watering and mixed type watering ($\chi^2$ = 194.02; p<0.001; $\chi^2$ = 17.76; p<0.001 respectively). The statistically significant higher sero-prevalence with regard to breeding method were observed only between herds utilizing all other breeding methods (own male, mixed methods, common use male) and those utilizing a male from another farm ($\chi^2$ = 18.57;p<0.001, $\chi^2$ = 26.73;p<0.001, $\chi^2$ = 27.04;p<0.001 respectively) as well as those utilizing AI ($\chi^2$ = 4.61; p = 0.032), $\chi^2$ = 6.77;p = 0.009, $\chi^2$ = 7.27;p = 0.007 respectively).

## Association between FMD sero-positivity and selected potential risk factors

Pairwise Spearman correlation of all the potential risk factors in Tables 3 and 4 showed significant moderate to strong correlation between many factors except age, sex, production zone, whether herds brought in SR, production type, breeding method and elevation. These were retained in the group of potential risk factors for FMD sero-positivity risk factor analysis. The most parsimonious mixed effects logistic regression model showing the association between FMD sero-positivity in small ruminants and risk factors as well as the relevant interactions are in Table 5.

In this final model obtained, the AIC was 7079.6 and the log likelihood was -3532.8 which indicated good fit for the data.

Only multipurpose production type showed statistically significant positive association when compared with meat production type. Thus multipurpose production type was 1.307 times more likely to be associated with FMD sero-positivity when compared with meat production type (p = 0.042). Interpretation of OR for risk factors that were negatively associated with FMD sero-positivity was after finding the inverse of OR (1/OR) as specified by Bland and Altman [55]. Therefore with reference to female animals, male animals were 1.238 times less

**Table 5. Association between FMD sero-positivity in small ruminants and risk factors studied, Kenya, 2016.**

| Risk factor | Variable | p | OR | 95%CI of OR |
|---|---|---|---|---|
| Intercept | Included | <0.001 | 2.459 | 2.047–2.872 |
| Sex | Female | Ref | | |
| | Male | 0.008 | 0.808 | 0.650–0.966 |
| Age | Mature | Ref | | |
| | Young | <0.001 | 0.291 | -0.420–1.002 |
| Production zone | Pastoral | Ref | | |
| | Sedentary | <0.001 | 0.278 | -0.225–0.780 |
| Production type | Meat | Ref | | |
| | Multipurpose | 0.042 | 1.307 | 1.049–1.566 |
| | Mixed | 0.608 | 0.876 | 0.373–1.380 |
| | Dairy | 0.416 | 1.351 | 0.626–2.076 |
| Sex*age interaction | Male*young | 0.019 | 3.671 | 3.671–4.754 |
| | Mature Female versus young female | 0.004 | 3.436 | 2.725–4.147 |
| | Mature female versus mature male | 0.041 | 1.238 | 1.079–1.396 |
| | Mature female versus young male | 0.986 | 1.158 | 0.322–1.995 |
| | Young female versus mature male | 0.029 | 0.360 | -0.363–1.084 |
| | Young female versus young male | 0.193 | 0.337 | -0.735–1.409 |
| | Mature male versus young male | 0.999 | 0.936 | 0.090–1.782 |

likely to be seropositive for FMD (p = 0.008). Compared to mature animals, young animals were 3.436 times less likely to be seropositive for FMD (p = <0.001) Animals in the sedentary zone were 3.597 times less likely to be sero-positive when compared with those in the pastoral zone (p<0.001). County and village IDs (with village IDs nested within counties) were used in the model as random effects variables but both effects were non-existent. The variances and Sd associated with observations within a village and county were 1.11 (Sd = 1.05) and 0.41 (Sd = 0.67), respectively. Scaled residuals ranged between -2.30 and 5.29. A model used to investigate interactions showed that interaction between age and sex was significant (p = 0.019). An animal that was mature and female was 3.436 times more likely to be sero-positive in contrast to being young and female (p = 0.004) and 1.238 times more likely to be sero-positive in contrast to being mature and male (p = 0.041). An animal that was young and female was 2.78 (1/0.360) times less likely to be sero-positive in contrast to one that was mature and male (p = 0.029). Thus an interaction with matureness or femaleness increased the risk of FMD sero-positivity above that for sex or age alone respectively.

## Discussion

The mean SR herd sizes in the PZ and SZ were 27.5 and 2.7 respectively. This is consistent with what has been reported in sub-saharan sedentary production systems [58]. The herd structure in the pastoral zone is similar to what has been reported in Somalia [59]. For the PZ, this is within the range reported recently in Kenya [38] but lower than that reported by Zaal [37], probably due to dwindling land available for livestock keeping and other changes in farming systems. According to this study, the bulk of SR in Kenya are held in the PZ.

Only 7564 samples from 872 herds were available for testing for FMD sero-prevalence compared to a calculated sample size of 9044. Though slightly lower than the calculated sample size, due to sample loss and lack of usability of some samples for the test (16%), these samples were deemed sufficient for determination of the sero-prevalence of FMD in the SR herds given

that there was sufficient design effect (2) consideration and provision for sample loss in sample size calculation.

The country sero-prevalence of FMD in SR was found to be 22.5% similar to what has been reported in other countries where FMD is endemic [16, 17]. It is however higher than that reported in Ethiopia, Israel, Libya and Sudan [8, 9, 11, 12, 18] but about half of what has been reported in Tanzania and Myanmar [14, 19, 20]. A previous study in cattle in Kenya showed much higher sero-prevalence in cattle at 52.5% [60] and unpublished data obtained at the same time with this current study in Kenyan cattle revealed a sero-prevalence of 37.6%. This means sheep and goats in Kenya could be less susceptible to FMDV compared to cattle despite the fact that they are normally herded together in endemic settings of Kenya as was also observed in Ethiopia [8, 9].

In the absence of vaccination, sero-prevalence to FMDV can be an indicator of presence of FMD. Sero-prevalence was significantly higher in the PZ (31.5%) than in the SZ (14.5%). This may be attributed to a high level of herd mobility, contact of animals at grazing and watering points, dynamism of herds (frequent additions) and frequent contact with the livestock of neighbouring countries through cross-border contact in the PZ. These animals move across the boundaries for grazing, watering and also for illegal trade thus promoting the concept that FMD outbreaks are associated with animal movement. In the process of movement they also come in contact with other animals from different areas which are an important factor for the transmission of the disease. The livestock in pastoral areas also end up in some sedentary zones during the dry season, potentially spreading disease [61]. Foot and mouth being a disease spread due to movement of animals closer together makes sedentary zone have lower incidence of spread between herds. This is important because most of the SR are in the PZ where they are more often herded together with cattle and cross-transmission may be the reason for the observed sero-positivity. The sero-prevalence in counties within the PZ or bordering the PZ such as Lamu were significantly higher than those in the counties in the SZ as also reported by others [60, 62]. This might be due to differences in the movement and distribution of livestock, the level of contact between herds and ungulate wildlife, proximity to stock routes, the grazing patterns and watering sources in each county.

A significant difference was observed in sero-prevalence of FMD among mature (23.6%) and young sheep and goats (10.1%). This is in agreement with the results of others [19, 20] although the sero-positivity levels in our study were lower. The difference in sero-positivity between age groups may be due to the fact that mature animals may have experienced more exposures to FMD at grazing, watering point and at market than in age group less than one year. Therefore, adult animals might have acquired infection from multiple strains and serotypes thus producing antibodies against multiple virus incursions of FMD. It could also be due to cumulative sero-positivity through repeated infection in their longer life time. The low prevalence in young animals may also be indicative of persistent passive immunity and less frequency of exposure of the animal to the disease as the farmers keep their lambs and kids in the homesteads. Females showed higher sero-prevalence at 23.8% than males (21.9%). However, these results are in contrast to Ethiopian studies where 15.7% and 8.3% seroconversions were reported in male and female animals respectively [63] and 8.9% in female while 3.0% in male [8]. Sero-prevalence was significantly higher in the multipurpose production type than in all the other production types (meat, mixed, dairy). This may be possible because this production type is found mainly among pastoral and agro-pastoral systems where purchase of animals is from the market or middlemen and which in each case had high sero-prevalence. Other researchers have demonstrated higher FMD sero-prevalence in production types resembling the multi-purpose production type than in meat, mixed and dairy production types [11, 20] However, Mesfine et al. [8] in Ethiopia have demonstrated lower sero-prevalence in pastoral

and agro-pastoral production systems than in sedentary production systems. Further, there was high sero-prevalence in animals whose production type was not identified hence the need for further investigation of sero-prevalence between the production types. The proportion of herds with at least one animal sero-positive for FMD (herd prevalence) was high at 77.6% similar to 74.7% reported by Ahmed et al [41] in Ethiopian cattle but higher than that in the reports of Megersa et al [64] in southern Ethiopian cattle and Hussain et al [65] in Omani cattle with sero-prevalence of 48.1% and 55.2%, respectively. This comparison of SR herds and cattle herds is relevant given possible transmission from cattle to SR. The high prevalence of FMD at the herd level in our study might be due to the common practice of communal grazing and watering in the study area as was also alluded to by Ahmed et al [41].

In spite of many variables showing differences in proportions of seropositive animals across the categories, only multipurpose production type showed a statistically significant positive association with FMD sero-positivity. Male sex, young age and sedentary production zone showed a statistically significant negative relationship. Thus some husbandry related variables showed significant relationship with sero-positivity as has also been alluded to by Balinda et al. [21] in Uganda. These results demand for risk-based surveillance which considers the significant risk factors. They also call for extension services and policies for small ruminant keepers to advice on interventions and husbandry practices which could limit the circulation of FMDV among SR herds which could also reduce cross-transmission with cattle herds.

Vaccination of small ruminants against FMD in Kenya is non-existent due to scarce vaccine and cost implications [34]. It may be worthwhile to vaccinate SR in some scenarios, given the identified risk factors. The possibility of transmission of FMDV from cattle to SR needs to be researched.

## Limitations of the study

Testing was not done for circulating virus (e.g. virus isolation/PCR), and therefore it is unknown when these animals became sero-positive. Indeed, further investigations (potentially using the same samples) could be done and also to identify the serotypes that the SR were sero-positive to, using serotype specific ELISAs. There was significant sero-positivity in animals where some variable levels were unidentified hence the need to investigate further the level of sero-positivity with regard to these variables. Although this study was at national level, The results in some counties are useful in making conclusions about the sero-positivity of FMD in SR as there was sufficient sample size for the counties. However, in some counties the sample size was too small to make any meaningful conclusions and therefore planned studies with sufficient sample sizes are required.

## Conclusion

The bulk of small ruminants in Kenya are held in the pastoral zone. Small ruminant FMD sero-positivity established in most counties countrywide shows that FMD may be present in the species in majority of herds but this needs to be authenticated through isolation of the FMD virus. The study has shown that the sero-prevalence in small ruminants in Kenya is estimated at 23.3% with sheep and goats having almost equal sero-prevalence. Given the near ubiquitous distribution of sero-prevalence, it is possible that the FMD virus may be circulating in a significant proportion of closed SR herds. There is also possible cross transmission of FMDV across the species which needs investigation. The pastoral zone had higher sero-positivity as compared to the sedentary zone. This shows the importance of concentrating control efforts in the pastoral zone where sero-positivity is high but without neglecting the sedentary areas which usually suffer the highest production and productivity losses in case of FMD

outbreaks. Besides, livestock in the pastoral zone also end up in some counties in the sedentary zone during the dry season. Past efforts for control of FMD in Kenya centered on compulsory vaccination of cattle in areas mostly located in the sedentary areas. The findings of this study should be considered in the development of FMD risk-based surveillance and control plans in small ruminants alongside those of the cattle population with due consideration of the established risk factors. More risk factors should be identified through planned studies.

## Supporting information

**S1 Table. FMD sero-prevalence and associated risk factors in SR in various countries.** (DOCX)

**S2 Table. Number of sheep and goats in the herds sampled in the study area, Kenya, 2016.** (DOCX)

**S3 Table. Small ruminant FMD sero-positivity per county, Kenya, 2016.** (DOCX)

**S1 File. Data collection tools.** (DOCX)

## Acknowledgments

We acknowledge approval of the study by the Directorate of Veterinary Services (DVS), Kenya and respective county governments. The survey teams and Foot- and -mouth Disease Laboratory staff are acknowledged for sample and data collection as well as sample testing. The respective county livestock keepers are acknowledged for presentation of animals and provision of data. The DVS data entry team which comprised of Ruth Manasse, Peninah Khan and Nelly Achieng' are also acknowledged. We acknowledge the assistance of M/s Jane Poole of the International Livestock Research Institute in statistical analysis of data.

## Author Contributions

**Conceptualization:** Eunice C. Chepkwony, George C. Gitao, Gerald M. Muchemi, Abraham K. Sangula, Salome W. Kairu-Wanyoike.

**Data curation:** Eunice C. Chepkwony, Gerald M. Muchemi, Salome W. Kairu-Wanyoike.

**Formal analysis:** Eunice C. Chepkwony, Gerald M. Muchemi, Salome W. Kairu-Wanyoike.

**Funding acquisition:** Eunice C. Chepkwony, Salome W. Kairu-Wanyoike.

**Investigation:** Eunice C. Chepkwony, Salome W. Kairu-Wanyoike.

**Methodology:** Eunice C. Chepkwony, George C. Gitao, Gerald M. Muchemi, Abraham K. Sangula, Salome W. Kairu-Wanyoike.

**Project administration:** Salome W. Kairu-Wanyoike.

**Resources:** Eunice C. Chepkwony, Salome W. Kairu-Wanyoike.

**Supervision:** George C. Gitao, Gerald M. Muchemi, Abraham K. Sangula, Salome W. Kairu-Wanyoike.

**Validation:** Eunice C. Chepkwony, George C. Gitao, Gerald M. Muchemi, Abraham K. Sangula, Salome W. Kairu-Wanyoike.

**Visualization:** Eunice C. Chepkwony, George C. Gitao, Gerald M. Muchemi, Abraham K. Sangula, Salome W. Kairu-Wanyoike.

**Writing – original draft:** Eunice C. Chepkwony, Salome W. Kairu-Wanyoike.

**Writing – review & editing:** Eunice C. Chepkwony, George C. Gitao, Gerald M. Muchemi, Abraham K. Sangula, Salome W. Kairu-Wanyoike.

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
