## [Decision Letter · Decision Letter 0]

6 Aug 2020

PONE-D-20-15166

Epidemiological study on foot-and-mouth disease in small ruminants: sero-prevalence and risk factor assessment in Kenya

PLOS ONE

Dear Dr. Kairu-Wanyoike,

Thank you for submitting your manuscript to PLOS ONE. After careful consideration, we feel that it has merit but does not fully meet PLOS ONE’s publication criteria as it currently stands. Therefore, we invite you to submit a revised version of the manuscript that addresses the points raised during the review process.

Details on the sampling procedure, sample size and descriptive analysis the number of animals that were vaccinated for FMD and those with clinical signs are lacking. Several parts of introduction, results and discussion are repeated throughout the text.

We look forward to receiving your revised manuscript.

Kind regards,

Jagadeesh Bayry, DVM, PhD, HDR

Academic Editor

PLOS ONE

Journal Requirements:

2. In your Methods section, please provide additional location information of the study sites, including geographic coordinates for the data set if available.

3. We note that Figure1, 2, 3 in your submission containmap images which may be copyrighted. All PLOS content is published under the Creative Commons Attribution License (CC BY 4.0), which means that the manuscript, images, and Supporting Information files will be freely available online, and any third party is permitted to access, download, copy, distribute, and use these materials in any way, even commercially, with proper attribution. For these reasons, we cannot publish previously copyrighted maps or satellite images created using proprietary data, such as Google software (Google Maps, Street View, and Earth). For more information, see our copyright guidelines: http://journals.plos.org/plosone/s/licenses-and-copyright.

3.1.    You may seek permission from the original copyright holder of Figure1, 2, 3 to publish the content specifically under the CC BY 4.0 license.

3.2.    If you are unable to obtain permission from the original copyright holder to publish these figures under the CC BY 4.0 license or if the copyright holder’s requirements are incompatible with the CC BY 4.0 license, please either i) remove the figure or ii) supply a replacement figure that complies with the CC BY 4.0 license. Please check copyright information on all replacement figures and update the figure caption with source information. If applicable, please specify in the figure caption text when a figure is similar but not identical to the original image and is therefore for illustrative purposes only.

Reviewers' comments:

Reviewer's Responses to Questions

**Comments to the Author**

1. Is the manuscript technically sound, and do the data support the conclusions?

Reviewer #1: Partly

Reviewer #2: Yes

2. Has the statistical analysis been performed appropriately and rigorously? 

Reviewer #1: No

Reviewer #2: Yes

3. Have the authors made all data underlying the findings in their manuscript fully available?

Reviewer #1: Yes

Reviewer #2: Yes

4. Is the manuscript presented in an intelligible fashion and written in standard English?

Reviewer #1: Yes

Reviewer #2: Yes

5. Review Comments to the Author

Reviewer #1: Comments of the authors

General comments

The study on ‘Epidemiological study on foot-and-mouth disease in small ruminants: sero-prevalence and risk factor assessment in Kenya’ is a good and useful study that documented the prevalence of the disease at national level and identified risk factors of the disease that can assist risk based surveillance and control intervention in Kenya. The study has collected quite large and fairly representative sample of the country’s small ruminant population that could generate reliable results. However the manuscript needs improvement in several ways and the following comments are forwarded to improve the manuscript

The introduction is very long. Eight page introduction for research article is unusual.

There is lack of clarity in in the sampling procedure and sample size determination. The sampling procedure is not clear. Specially the term ‘herd’ was defined in different ways and used inconsistently. This made difficult to understand how the multistage sampling method was used. It was not also clear why the sample size was determined independently for the two zones (PZ and SZ). The same inputs (parameters) were used in each zone and the same number of herds and animals were taken from each zone. If the interest was to stratify the sample, the nationally determined sample size could have been divided among the zones.

The risk factor analysis was done using chi-square, and bivariable (I would suggest to name this univariable) and multivariable logistic regressions. If univariable logistic regression is done, the chi-square analysis is redundant and anything about chi-square in the manuscript should be removed. Looking in to that analysis even the univariable logistic regression is not important as it has not been used for screening the potential risk factor for the multivariable analysis. For that matter, given the adequate sample size, screening the variables is not needed and the univariable analysis can be ignored. Another problem with the analysis was; while the sampling procedure is cluster sampling, the analysis did not consider the sampling procedure. If cluster sampling is used there will underestimation of standard error (unwarranted significant p-values for regression coefficients) and this has to be taken care of. I would suggest use of mixed effect logistic regression with ‘herds’ and ‘villages’ as random effect variables for more reliable identification of factors associated with the disease.

The result has unnecessary detail and redundant results. In line with the comments given in statistical analysis above, the chi-square test and the univariable analysis provides the same result and there is no any need to do the chi-square analysis. The univaraible logistic regression give more information like crude odds ratio for each category of the categorical variables which is not directly possible in the chi-square analysis. So I suggest anything related the chi-square analysis. Even the importance of the univariable analysis result should be re-considred.

The discussion of the risk factor should be based on the significance of risk factors in the final multivariable model not on crude association seen from the univariable analysis.

All the conclusions should be supported by the study findings which was not the case for some the recommendations

The manuscript also needs improvement in the English.

Specific Comments

Abstract

The abstract followed unstructured format. In this type format the abstract should be written in one paragraph.

Line 33-32. No need of mentioning the statistical software in the abstract.

Line 34-5. If logistic regression is done, the use chi-square for risk factor analysis will be redundancy

Line 39 – 4 . Re -write it as “ the risk factors that were significantly positively associated with FMD sero-positivity in SR were being multipurpose (OR=1.150; p=0.034) and dairy (OR=2.029; p=0.003) production types.”

Line 51…’carrier SR’…... Subclinical carrier? You didn’t have any result that showed SR are acting as carriers FMD.

Introduction

- The introduction is very long it has to be shortened to maximum of not more than 2-3 pages. E.g. the extensive discussion about differential diagnosis and the different diagnostic assays can be removed. The extensive description of the small ruminant production system in Kenya can be shorted and taken to material and methods part. The extensive discussion about the seroprevalence of SR FMD and production systems worldwide can be shortened or removed.

Line 154. They ARE……

Line 166. Average HERD SIZE of sheep and goats

Line 176. .

Line 183. Is FMD important in sheep and goat as cause of production loss or for its epidemiological contributions for cattle? Just compare this with next paragraph (line 186-19)

Lin 2011-15. Do these studies support your claim that trade costs are more important than direct cost of FMD for households? Do these SM keeper households participate international trade to countries that free of the disease?

Line 226-230. The two sentences seem contradictory

Materials and methods

Line 248-9. ……………...since FMD is a transboundary disease and also transmitted through export of animals and animal products.

Line 255-57. ‘The study targeted the sub location’ this is not clear. In epidemiology target population has its own technical meaning. The target population in this study is the national SR population. Rephrase this sentence to write what you mean.

Line 252-7. Although the section title is study area, the text in this paragraph is more about study population

Line 283. Sample size calculation was in two stages and per zone) IN EACH STRATUM:

Line 286. Make clear what is in-contact farm

Line 287 …………… “assuming a simple random sample of herds in each stratum independently” not clear

Line 294 …………’simple random sample” SIMPLE RANDOM SAMPLING? But two line down it describes a cluster sampling in which first sub-locations are selected then household herds and then animals?

Line 297. ‘323 sub-locations’. The 323 were herds which were defined as farms/in contact farms not sub-location! Again on next line it says “one village (herd)” and ‘”household herd per village”. Please make clear what herd is and use it consistently; not “herd” one time and “household herd’ another time unless they are meant different things.

If the interest is to do stratified sampling the approach could have been determining the sample size using one of the sampling techniques (looks cluster sampling in this study case) and allocating the sample among the strata proportionally or if there reason not to allocate proportionally use other method of allocation. I couldn’t see the need to determine the sample size for the two strata independently as all the parameter used are the same for each stratum.

Line 243. What are field freezers?

Line 343 -344 . ‘One aliquot was used to test for the presence or absence of antibodies of Rift Valley Fever (RVF) and Peste de Petits Ruminants (PPR) antibodies according to the objective of the STSD project’ Give some explanation about this work and make clear that the present FMD work is a part or accompany of that work.

Line 363-63. Rewrite the sentence avoiding repetitions and put the right reference for the test kit (product, Manufacture Company and place/country).

Line 365-67 On the seropositivity/negativity to FMDV antibodies the outcome variables were categorised based on the on the results of the 3ABC blocking enzyme-linked immunosorbent assay.

Line 367-73. The test procedure is not clearly documented. Revise the English.

Line 390-94. Why was the herd level variables first entered to MS access (unlike the animal level variables which are directly entered to MS excel) before being brought to MS excel for data cleaning and coding?

Line 402- To do this you need to document the se and sp of the test in the laboratory diagnosis section.

Line 403- 433 statistical analysis part has lot of repetition and unnecessary detail. For example lin4 405-6 has the same idea with line 402-405 both of them are about crude association using chi-square. But few lines down it mention bivariable analysis is done. The chi- square test of comparison and bivariable analysis are the same and the chi-square is not needed for variables in considered in the bivariable logistic regressions

The terminology bivariable is not correct has to be changed into univariable. The common usage is univariable and multivariable. It about the number of independent variables. If only one independent variable is included it is univariable and if more than one indnepenet variables is included it is multivariable.

Line 434-36 The results of our study have been presented mainly in tables and figures and interpreted in text. Although the bivariable regression was carried out for all risk factors (individual animal and herd level) together, the results are in two tables to avoid too large a table.

Other issues in this section;

- The purpose the univariable analysis should be mentioned.

- For goodness of it test, it is enough to mention what tests are used with appropriate reference. How each test measures goodness of fit is unnecessary detail.

- In several places in this part the word’ interpretation’ is incorrectly used. For example specifying confidence interval level and P- values (419-20) or putting results in text (line 434-35) are not interpretations. Interpretations ae given meaning for you findings that is done in the discussion.

Results

Line 450. In M&M, the number of herds sampled was stated as 323 herds* 2 strata = 646 so how the herd number increased to 898.

Line 451. Mention why the remaining samples were not available for testing

Line 472. However a large proportion ……

Line 482. Write in the full ideas instead of using “so” to make it clearer

Line 496. What is the diffidence between communal grazing and mixed grazing

Line 515-16. Sero-prevalence of a total of 7564 sera from the whole country was determined for the

presence of non-structural FMDV protein (antibodies).

Line 517. “Prevalence rate” Prevalence is not rate. Simply use prevalence (which known to be proportion). Correct this throughout the document.

Line 518. if χ2 has to be reported it has to include the degrees of freedom like χ2(3), which means χ2 at 3 degrees of freedom. Do this throughout the manuscript

Line 516. Write it as apparent prevalence as you have also true prevalence estimate

Line 527. ‘Showed seronegativity’ is not good expression; state that prevalence in these counties is zero

Line 530. Put the estimated true Seroprevalence in figure.

Line 531-535. What does samplings sites represent, villages, individual herds or what? the map in figure 3 does not provide any information except the distribution of sampling sites. It should have indicate the negative sites as well. Moreover the legend and labels in the maps are not visible.

Line 539 PAIR WISE Pearson correlation…………

Line 539 …‘between county and production zone”….. No risk factor variable called ‘county type’ was mentioned in the list of variables indicated M&M part. Moreover test of collinearity is needed for the multivariable analysis. So this sentence should go down where multivariable results are documented. Another issue here is the use of the term ‘risk factor” should be replaced by “potential risk factor”. The variables are hypothesized to be a risk factor, not yet confirmed to be risk factor. They will be risk factor when they are found so after the analysis. So they should be referred to as potential risk factors thought out the document until they are confirmed.

Line 546-47. Where is the result of the statistical analysis i.e. P-value

Line 546-53. In this section it seems χ2 is used but in chi-square for comparison of three or more groups, it tells you only whether there is significant difference in the proportion/ prevalence among all groups. It doesn’t tell you between which pairs is the difference significant. So how can you come up with this results where the variables has three categories like breed (exotic, local, cross)?

Line 554. Replace ‘the between herd prevalence’ with ‘herd level prevalence”

Line 564. If it is not significant, why is it surprising? In the first place, in the result section just put what you find; such interpretation should be reserved for the discussion

Line 566-71. The same comment as above in line 546-53.

Line 604- 5. The result emphasis should be which factors are significantly associated with FMD seropositivty. Whether the association is positive or negative depends how the data was coded during analysis. For example, it is not sex that is negatively associated but it is being maleness.

Line 604. Instead of table 5, S4 and S5 should be part of the main manuscript not supplementary.

Line 612. S5 table attached is not about the herd level variables. Check whether the attached table is the right one.

Line 612-13. The comparator should be mentioned.

Line 627-8. The purpose of univaraible analysis is to screen the potential risk factors. If cut off p-value 0.2 does not reduce them small cut off should have been used. Actually given the very large sample size there was no need to screen in this study and univaraible analysis was not necessary.

Line 645-76. Repetition of the table. Enough to mention the most salient results and leave the remaining detail to be referred from table

Discussion

Line 685. Delete ‘It is true’ and start with ‘According to this study …..’

Line 724-30. For SR age of 6 months and above can be considered as matured as they can be bred at this age and classification of mature above 1years of age is not appropriate. Hence it difficult to think these age animals (age 6 months – 1 year) will be kept at homestead and are less exposed to the disease. The observed higher prevalence in older animals could also be due cumulative seropositivity through repeated infection in their longer life time.

Line 730. conversely

Line 731. ‘Female posted’ change the word ‘posted’ with other more appropriate word

Line 730-32. Both the univariable and the multivariable analysis showed females have significantly higher prevalence than males. Which result is being discussed here!

Line 740 ‘….position…’ to mean explanation?

Line 745. Change the word ‘contrary’ with the actual type purchase it self

Line 749-50. not clear

Line 755-59. Make it clearer

Line 768. … three..

Line 784-95. Merge this discussion with preceding paragraphs. In the first place, what should be discussed is the result of the final model where confounding is controlled. It is meaningless to discuss the crude results of the chi-square or univariable analysis as these can’t correctly identify significant risk factors because of confounding. So the discussion in the preceding paragraphs should be adjusted accordingly.

Conclusion

Line 820-21. “The findings of this study give an understanding of the potential role of small ruminants in the epidemiology of FMD in Kenya and contribute to the global scenario’ Not exactly. It is difficult to say anything about the role of SR in epidemiology of FMD based on this study.

Line 824-25. You found high Seroprevalence does not confirm that they transmit and maintain the disease. So this conclusion cannot be warranted based on your findings.

Line 826. “Cross contamination” has to be replaced with cross transmission

Line 827. ‘This outlays’?

Line 829 ..”post”.. ??

Figures

The legends and labels of all figures (fig.1,2,3) are not visible

Reviewer #2: This study aims to determine the seroprevalence of FMD in small ruminants and determine the risk factors associated. This is an original, highly useful study that may be utilised to further understand the epidemiology of FMD transmission in Kenya.

The manuscript is very thorough, however is much too long, particularly the introduction. The whole manuscript needs to be made more concise, as there is a lot of repetition throughout, between sections, between the text and figures, and between figures. Some tables could be merged, with some columns removed – for example tables 3 and S4. The current layout makes it more confusing to the reader than it needs to be – as this is a simple study approach with good results. However, the message seems to get lost in the length of the manuscript.

For the introduction, only information relevant to the study should be included, for example detail on molecular tests and differential diagnosis is not required – potentially only mentioned. Much of the information is or could be included in the discussion if it is necessary. Additionally, please ensure that you include references where appropriate, there are multiple sections where there are not enough references.

The methods section is very repetitive, particularly regarding the description of the number of counties, and the data analysis section could be much more concise. Additionally, the ELISA method does not need to be described in such detail as it is a commonly used test.

In the results section, please ensure that your subheadings accurately reflect what is included within them and avoid repetition between tables and the text. Additionally, are chi-squared test required if regression has been done? Or can this be more concisely displayed e.g. all in one table? At the moment it is very difficult to flip between all of the tables and the lengthy text to work out the results.

Additionally, I would like to see in the descriptive analysis the number of animals that were vaccinated for FMD and that had clinical signs (even if it was 0). This is very important information when investigating seroprevalence. Particularly when the vaccines administered in Kenya are often not of high quality and may induce NSP antibodies – this should also be mentioned in the discussion.

Additionally, please be careful with capitalised words that do not need to be, for example ‘county’ and directions (east/west etc.)

Consequently I recommend major revisions to this manuscript to make it more 'reader friendly'.

6. PLOS authors have the option to publish the peer review history of their article (what does this mean?). If published, this will include your full peer review and any attached files.

Reviewer #1: No

Reviewer #2: **Yes: **Bryony Armson

---

## [Author Response · Author response to Decision Letter 0]

24 Feb 2021

Salome Kairu-Wanyoike,

Meat Training Institute,

Directorate of Veterinary Services,

State Department of Livestock,

P.O. Box 55-00204, 

Athi-River, Kenya.

February 5 2021

The Editor

PLoS ONE Journal

Dear Editor,

I write to submit the corrected manuscript “Epidemiological study on foot-and-mouth disease in small ruminants: sero-prevalence and risk factor assessment in Kenya” as a full research article to PLoS ONE. 

All the authors reviewed this corrected manuscript and agree to the submission. There are no opposed reviewers. We very much appreciate the opportunity that has been offered to us to correct the manuscript and the valuable inputs by the reviewers. Appended is the response to reviewers on issues raised.

Sincerely, 

 Salome Kairu-Wanyoike, Ph.D. 

 2. In your Methods section, please provide additional location information of the study sites, including geographic coordinates for the data set if available.

 Provided in an updated file now S2 file

3. We note that Figure1, 2, 3 in your submission containmap images which may be copyrighted. All PLOS content is published under the Creative Commons Attribution License (CC BY 4.0), which means that the manuscript, images, and Supporting Information files will be freely available online, and any third party is permitted to access, download, copy, distribute, and use these materials in any way, even commercially, with proper attribution. For these reasons, we cannot publish previously copyrighted maps or satellite images created using proprietary data, such as Google software (Google Maps, Street View, and Earth). For more information, see our copyright guidelines: http://journals.plos.org/plosone/s/licenses-and-copyright.

Figures 2 and 3 were produced as part of the project work by part of the authors of this manuscript who were the principal investigators in the project. The figures may however be in project reports and documents which may appear online elsewhere. Figures 1 and 3 have been removed. Figure 2 has been retained to give a pictorial overview of the study zones and sampling sites.

Reviewers' comments:

Reviewer's Responses to Questions

Comments to the Author

1. Is the manuscript technically sound, and do the data support the conclusions?

Reviewer #1: Partly

This has been addressed.

Reviewer #2: Yes

2. Has the statistical analysis been performed appropriately and rigorously?

Reviewer #1: No

This has been addressed.

Reviewer #2: Yes

3. Have the authors made all data underlying the findings in their manuscript fully available?

Reviewer #1: Yes

Reviewer #2: Yes

4. Is the manuscript presented in an intelligible fashion and written in standard English?

Reviewer #1: Yes

Reviewer #2: Yes

5. Review Comments to the Author

Reviewer #1: Comments of the authors

General comments

The study on ‘Epidemiological study on foot-and-mouth disease in small ruminants: sero-prevalence and risk factor assessment in Kenya’ is a good and useful study that documented the prevalence of the disease at national level and identified risk factors of the disease that can assist risk based surveillance and control intervention in Kenya. The study has collected quite large and fairly representative sample of the country’s small ruminant population that could generate reliable results. 

Compliments appreciated

However the manuscript needs improvement in several ways and the following comments are forwarded to improve the manuscript

The introduction is very long. Eight page introduction for research article is unusual.

The introduction has been reduced considerably.

There is lack of clarity in in the sampling procedure and sample size determination. The sampling procedure is not clear. Specially the term ‘herd’ was defined in different ways and used inconsistently. This made difficult to understand how the multistage sampling method was used. It was not also clear why the sample size was determined independently for the two zones (PZ and SZ). The same inputs (parameters) were used in each zone and the same number of herds and animals were taken from each zone. If the interest was to stratify the sample, the nationally determined sample size could have been divided among the zones.

These have been clarified in the various sections.

The risk factor analysis was done using chi-square, and bivariable (I would suggest to name this univariable) and multivariable logistic regressions. If univariable logistic regression is done, the chi-square analysis is redundant and anything about chi-square in the manuscript should be removed. Looking in to that analysis even the univariable logistic regression is not important as it has not been used for screening the potential risk factor for the multivariable analysis. For that matter, given the adequate sample size, screening the variables is not needed and the univariable analysis can be ignored. 

Chi-square analysis and univariable regression analysis for test of association have been removed. The only Chi-square analyses remaining in the manuscript are for the test of difference in proportions.

Another problem with the analysis was; while the sampling procedure is cluster sampling, the analysis did not consider the sampling procedure. If cluster sampling is used there will underestimation of standard error (unwarranted significant p-values for regression coefficients) and this has to be taken care of. I would suggest use of mixed effect logistic regression with ‘herds’ and ‘villages’ as random effect variables for more reliable identification of factors associated with the disease.

Generalized linear mixed effects logistic regression models have been run with county and villages as random effect variables. Interactions between fixed effect variables has also been tested.

The result has unnecessary detail and redundant results. In line with the comments given in statistical analysis above, the chi-square test and the univariable analysis provides the same result and there is no any need to do the chi-square analysis. The univaraible logistic regression give more information like crude odds ratio for each category of the categorical variables which is not directly possible in the chi-square analysis. So I suggest anything related the chi-square analysis. Even the importance of the univariable analysis result should be re-considered.

Removed as stated above.

The discussion of the risk factor should be based on the significance of risk factors in the final multivariable model not on crude association seen from the univariable analysis.

Chi-square and univariable analyses have been removed

All the conclusions should be supported by the study findings which was not the case for some the recommendations

This has been done

The manuscript also needs improvement in the English.

English has been corrected throughout the manuscript

Specific Comments

Abstract

The abstract followed unstructured format. In this type format the abstract should be written in one paragraph.

The abstract is now in one paragraph

Line 33-32. No need of mentioning the statistical software in the abstract.

Statistical software has been removed on Line 33/34

Line 34-5. If logistic regression is done, the use chi-square for risk factor analysis will be redundancy

Chi-square has been removed in Line 36

Line 39 – 40 . Re -write it as “ the risk factors that were significantly positively associated with FMD sero-positivity in SR were being multipurpose (OR=1.150; p=0.034) and dairy (OR=2.029; p=0.003) production types.”

Done in Line 39-41

Line 51…’carrier SR’…... Subclinical carrier? You didn’t have any result that showed SR are acting as carriers FMD.

The term ‘carrier’ has been removed on Line 57

Introduction

The introduction is very long it has to be shortened to maximum of not more than 2-3 pages. E.g. the extensive discussion about differential diagnosis and the different diagnostic assays can be removed. The extensive description of the small ruminant production system in Kenya can be shorted and taken to material and methods part. The extensive discussion about the seroprevalence of SR FMD and production systems worldwide can be shortened or removed.

The introduction has been shortened.

Line 154. They ARE……

Corrected on Line 67.

Line 166. Average HERD SIZE of sheep and goats

Corrected on Line 385 in materials and methods.

Line 176. .

Line 183. Is FMD important in sheep and goat as cause of production loss or for its epidemiological contributions for cattle? Just compare this with next paragraph (line 186-19)

Sentence has been removed.

Lin 2011-15. Do these studies support your claim that trade costs are more important than direct cost of FMD for households? Do these SM keeper households participate international trade to countries that free of the disease?

Sentence has been removed

Line 226-230. The two sentences seem contradictory

The two sentences have been removed.

Materials and methods

Line 248-9. ……………...since FMD is a transboundary disease and also transmitted through export of animals and animal products.

Corrected on Line 145-147

Line 255-57. ‘The study targeted the sub location’ this is not clear. In epidemiology target population has its own technical meaning. The target population in this study is the national SR population. Rephrase this sentence to write what you mean.

Rephrased on Line 341-342

Line 252-7. Although the section title is study area, the text in this paragraph is more about study population

A study population section has been created 

Line 283. Sample size calculation was in two stages and per zone) IN EACH STRATUM:

Included in Line 394

Line 286. Make clear what is in-contact farm

Clarified on Line 397-398

Line 287 …………… “assuming a simple random sample of herds in each stratum independently” not clear

The phrase has been deleted in Line 398-399

Line 294 …………’simple random sample” SIMPLE RANDOM SAMPLING? But two line down it describes a cluster sampling in which first sub-locations are selected then household herds and then animals?

Phrase has been removed in Line 405-406.

Line 297. ‘323 sub-locations’. The 323 were herds which were defined as farms/in contact farms not sub-location! Again on next line it says “one village (herd)” and ‘”household herd per village”. Please make clear what herd is and use it consistently; not “herd” one time and “household herd’ another time unless they are meant different things.

Clarified on Line 408-413. But note that because it was one village per sub-location and one herd per village, it is appropriate to use the terms interchangeably but one term has now been maintained for consistency. 

If the interest is to do stratified sampling the approach could have been determining the sample size using one of the sampling techniques (looks cluster sampling in this study case) and allocating the sample among the strata proportionally or if there reason not to allocate proportionally use other method of allocation. I couldn’t see the need to determine the sample size for the two strata independently as all the parameter used are the same for each stratum.

The cluster sampling was more appropriate given the administrative structure of the country. The two zones are quite distinct in structure even if FMD dynamics (in cattle) seem not very different. The SZ has small counties with very many sub-locations while the PZ has large counties with fewer sub-locations which could have led to high sampling in the SZ and low sampling in the PZ. It was also important to see the difference in the two zones and we chose to have a complete separation between the two and sample equal number of sub-locations in each. This is also explained in line 414-421.

Line 243. What are field freezers?

Clarified on Line 461

Line 343 -344 . ‘One aliquot was used to test for the presence or absence of antibodies of Rift Valley Fever (RVF) and Peste de Petits Ruminants (PPR) antibodies according to the objective of the STSD project’ Give some explanation about this work and make clear that the present FMD work is a part or accompany of that work. 

Explained in Line 465-468

Line 363-63. Rewrite the sentence avoiding repetitions and put the right reference for the test kit (product, Manufacture Company and place/country).

Done in Line 488-492

Line 365-67 On the seropositivity/negativity to FMDV antibodies the outcome variables were categorised based on the on the results of the 3ABC blocking enzyme-linked immunosorbent assay.

Corrected in Line 492-495

Line 367-73. The test procedure is not clearly documented. Revise the English.

The procedure has been revised in Line 488-496. The details on the test procedure have been removed as advised by reviewer 2 (Line 497-513)

Line 390-94. Why was the herd level variables first entered to MS access (unlike the animal level variables which are directly entered to MS excel) before being brought to MS excel for data cleaning and coding?

Herd level variables were first entered into MS Access because of the large amount of data which needed to be compartmentalized in tables which could then be linked as appropriate (happens better in MS Access) before export to MS Excel for analysis as needed. Animal level variables were few and easy to enter into MS Excel in one sheet. Besides animal level data were in the sampling forms in the custody of the laboratory personnel while the herd level data were in questionnaires in the custody of data entry personnel in the epidemiology department. Entering the data separately saved time as one did not have to wait for the other. Linking the two data sets after entry was easy. See Line 524-525

Line 402- To do this you need to document the se and sp of the test in the laboratory diagnosis section.

Done in Line 492

Line 403- 433 statistical analysis part has lot of repetition and unnecessary detail. For example line 405-6 has the same idea with line 402-405 both of them are about crude association using chi-square. 

Chi-square is used i) to test differences in proportions and ii) to test crude associations. The statement in Line 403-5 is about testing the difference in proportions and has been retained while that in Line405-6 is about testing the crude association between FMD sero-positivity and potential risk factors and has been removed in accordance with the general comments of the authors. The statistical analysis has been streamlined. See Line 539-541.

But few lines down it mention bivariable analysis is done. The chi- square test of comparison and bivariable analysis are the same and the chi-square is not needed for variables in considered in the bivariable logistic regressions

Removed as mentioned above

The terminology bivariable is not correct has to be changed into univariable. The common usage is univariable and multivariable. It about the number of independent variables. If only one independent variable is included it is univariable and if more than one indnepenet variables is included it is multivariable.

Corrected

Line 434-36 The results of our study have been presented mainly in tables and figures and interpreted in text. Although the bivariable regression was carried out for all risk factors (individual animal and herd level) together, the results are in two tables to avoid too large a table.

Sentences have been removed in Line 582-584

Other issues in this section;

- The purpose the univariable analysis should be mentioned.

Univariable analysis has been removed

- For goodness of it test, it is enough to mention what tests are used with appropriate reference.

Done in Line 571-573. How each test measures goodness of fit is unnecessary detail.

Unnecessary detail removed in Line 573-581

- In several places in this part the word’ interpretation’ is incorrectly used. For example specifying confidence interval level and P- values (419-20) or putting results in text (line 434-35) are not interpretations. Interpretations ae given meaning for you findings that is done in the discussion. 

Terms have been removed in Line 565-6 and 582-3.

Results

Line 450. In M&M, the number of herds sampled was stated as 323 herds* 2 strata = 646 so how the herd number increased to 898.

If one herd could yield all the animals required, only that herd was sampled, otherwise additional herds that were in contact with the selected herds were sampled until the required number to be sampled in the sub-location was reached as also mentioned in Line 435-437. See also Line 602-603.

Line 451. Mention why the remaining samples were not available for testing

Mentioned in Line 604-605

Line 472. However a large proportion ……

Corrected on Line 626

Line 482. Write in the full ideas instead of using “so” to make it clearer

Corrected on Line 636 and 637

Line 496. What is the difference between communal grazing and mixed grazing?

Explained in Line 655

Line 515-16. Sero-prevalence of a total of 7564 sera from the whole country was determined for the

presence of non-structural FMDV protein (antibodies).

Corrected on Line 671-673

Line 517. “Prevalence rate” Prevalence is not rate. Simply use prevalence (which known to be proportion). Correct this throughout the document.

Corrected throughout the document

Line 518. if χ2 has to be reported it has to include the degrees of freedom like χ2(3), which means χ2 at 3 degrees of freedom. Do this throughout the manuscript

Where chi-square test is for comparison of two proportions at a time as in this case, the degree of freedom is 1 and is usually not reported. The results of chi-square test for test of crude association where the degrees of freedom are higher have been removed from the manuscript as per the reviewer recommendations.

Line 516. Write it as apparent prevalence as you have also true prevalence estimate

Corrected on Line 673

Line 527. ‘Showed seronegativity’ is not good expression; state that prevalence in these counties is zero

Corrected on Line 684-685

Line 530. Put the estimated true Seroprevalence in figure.

Inserted in Line 687

Line 531-535. What does samplings sites represent, villages, individual herds or what? the map in figure 3 does not provide any information except the distribution of sampling sites. It should have indicate the negative sites as well. Moreover the legend and labels in the maps are not visible.

Figure 3 has been removed. Explanation is now only in text in Line 689-691

Line 539 PAIR WISE Pearson correlation…………

Corrected on Line 785

Line 539 …‘between county and production zone”….. No risk factor variable called ‘county type’ was mentioned in the list of variables indicated M&M part. Included in L524

 Moreover test of collinearity is needed for the multivariable analysis. So this sentence should go down where multivariable results are documented. 

Inserted in Line 785-788

Another issue here is the use of the term ‘risk factor” should be replaced by “potential risk factor”. The variables are hypothesized to be a risk factor, not yet confirmed to be risk factor. They will be risk factor when they are found so after the analysis. So they should be referred to as potential risk factors thought out the document until they are confirmed.

Done throughout the document.

Line 546-47. Where is the result of the statistical analysis i.e. P-value

Provided on Line 705-6

Line 546-53. In this section it seems χ2 is used but in chi-square for comparison of three or more groups, it tells you only whether there is significant difference in the proportion/ prevalence among all groups. It doesn’t tell you between which pairs is the difference significant. So how can you come up with this results where the variables has three categories like breed (exotic, local, cross)?

The comparisons were of two categories at any time like local with cross-breed; local with exotic; cross-breed with exotic and not all three at the same time. 

Line 554. Replace ‘the between herd prevalence’ with ‘herd level prevalence”

Done in Line 712

Line 564. If it is not significant, why is it surprising? In the first place, in the result section just put what you find; such interpretation should be reserved for the discussion

Corrected on Line 772

Line 566-71. The same comment as above in line 546-53.

Same explanation holds as for Line 546-53

Line 604- 5. The result emphasis should be which factors are significantly associated with FMD seropositivty. Whether the association is positive or negative depends how the data was coded during analysis. For example, it is not sex that is negatively associated but it is being maleness.

Corrected in Line 813-828

Line 604. Instead of table 5, S4 and S5 should be part of the main manuscript not supplementary.

All these tables have been removed from the manuscript.

Line 612. S5 table attached is not about the herd level variables. Check whether the attached table is the right one.

Table has been removed altogether

Line 612-13. The comparator should be mentioned.

The analysis has been removed

Line 627-8. The purpose of univaraible analysis is to screen the potential risk factors. If cut off p-value 0.2 does not reduce them small cut off should have been used. Actually given the very large sample size there was no need to screen in this study and univaraible analysis was not necessary.

Univariable analysis removed

Line 645-76. Repetition of the table. Enough to mention the most salient results and leave the remaining detail to be referred from table

Corrected 

Discussion

Line 685. Delete ‘It is true’ and start with ‘According to this study …..’

Corrected on Line 870

Line 724-30. For SR age of 6 months and above can be considered as matured as they can be bred at this age and classification of mature above 1years of age is not appropriate. Hence it difficult to think these age animals (age 6 months – 1 year) will be kept at homestead and are less exposed to the disease. 

Under Kenyan conditions, SR under one year are still not mature and are rarely bred. They remain close to the homestead or are grazed separate from the main herd.

The observed higher prevalence in older animals could also be due cumulative seropositivity through repeated infection in their longer life time.

Included in the discussion L917-918

Line 730. Conversely

Corrected on Line 920

Line 731. ‘Female posted’ change the word ‘posted’ with other more appropriate word

Corrected on Line 921

Line 730-32. Both the univariable and the multivariable analysis showed females have significantly higher prevalence than males. Which result is being discussed here!

Corrected on Line 920-923

Line 740 ‘….position…’ to mean explanation?

Removed

Line 745. Change the word ‘contrary’ with the actual type purchase it self

Removed

Line 749-50. not clear

Removed

Line 755-59. Make it clearer

Clarified on Line 945-952

Line 768. … three..

Removed

Line 784-95. Merge this discussion with preceding paragraphs. In the first place, what should be discussed is the result of the final model where confounding is controlled. It is meaningless to discuss the crude results of the chi-square or univariable analysis as these can’t correctly identify significant risk factors because of confounding. So the discussion in the preceding paragraphs should be adjusted accordingly.

Done

Conclusion

Line 820-21. “The findings of this study give an understanding of the potential role of small ruminants in the epidemiology of FMD in Kenya and contribute to the global scenario’ Not exactly. It is difficult to say anything about the role of SR in epidemiology of FMD based on this study.

Corrected in Line 1013-16

Line 824-25. You found high Seroprevalence does not confirm that they transmit and maintain the disease. So this conclusion cannot be warranted based on your findings.

Revised on line 1020

Line 826. “Cross contamination” has to be replaced with cross transmission

Replaced in Line 1021

Line 827. ‘This outlays’?

Corrected on Line 1022

Line 829 ..”post”.. ??

Corrected on Line 1024

Figures

The legends and labels of all figures (fig.1,2,3) are not visible

Fig. 1 and 3 have been removed and fig. 2 has been improved.

Reviewer #2: This study aims to determine the seroprevalence of FMD in small ruminants and determine the risk factors associated. This is an original, highly useful study that may be utilised to further understand the epidemiology of FMD transmission in Kenya.

The manuscript is very thorough, however is much too long, particularly the introduction. The whole manuscript needs to be made more concise, as there is a lot of repetition throughout, between sections, between the text and figures, and between figures. Some tables could be merged, with some columns removed – for example tables 3 and S4. The current layout makes it more confusing to the reader than it needs to be – as this is a simple study approach with good results. However, the message seems to get lost in the length of the manuscript.

For the introduction, only information relevant to the study should be included, for example detail on molecular tests and differential diagnosis is not required – potentially only mentioned. Much of the information is or could be included in the discussion if it is necessary. Additionally, please ensure that you include references where appropriate, there are multiple sections where there are not enough references.

The methods section is very repetitive, particularly regarding the description of the number of counties, and the data analysis section could be much more concise. Additionally, the ELISA method does not need to be described in such detail as it is a commonly used test.

In the results section, please ensure that your subheadings accurately reflect what is included within them and avoid repetition between tables and the text. Additionally, are chi-squared test required if regression has been done? Or can this be more concisely displayed e.g. all in one table? At the moment it is very difficult to flip between all of the tables and the lengthy text to work out the results.

Additionally, I would like to see in the descriptive analysis the number of animals that were vaccinated for FMD and that had clinical signs (even if it was 0). This is very important information when investigating seroprevalence. Particularly when the vaccines administered in Kenya are often not of high quality and may induce NSP antibodies – this should also be mentioned in the discussion.

Additionally, please be careful with capitalised words that do not need to be, for example ‘county’ and directions (east/west etc.)

Consequently I recommend major revisions to this manuscript to make it more 'reader friendly'. We appreciate the compliments regarding the study usefulness and thoroughness. The introduction has been shortened and irrelevancies removed. Repetitions have also been removed. S4 has been removed. Description of the ELISA method has been summarized. Chis-quare test of crude association results have been removed. In Kenya sheep and goats are not vaccinated against FMD as indicated in Line138 and 1015-16 No clinical signs were encountered during the survey as indicated now in Line 671. Unnecessary capitalization of words has been removed.

---

## [Decision Letter · Decision Letter 1]

7 Apr 2021

PONE-D-20-15166R1

Epidemiological study on foot-and-mouth disease in small ruminants: sero-prevalence and risk factor assessment in Kenya

PLOS ONE

Dear Dr. Kairu-Wanyoike,

Thank you for submitting your manuscript to PLOS ONE. After careful consideration, we feel that it has merit but does not fully meet PLOS ONE’s publication criteria as it currently stands. Therefore, we invite you to submit a revised version of the manuscript that addresses the points raised during the review process.

Overall revision was satisfactory. Both the reviewers have identified several minor points that we request you to consider for the revision.

We look forward to receiving your revised manuscript.

Kind regards,

Jagadeesh Bayry, DVM, PhD, HDR

Academic Editor

PLOS ONE

Journal Requirements:

Reviewers' comments:

Reviewer's Responses to Questions

**Comments to the Author**

1. If the authors have adequately addressed your comments raised in a previous round of review and you feel that this manuscript is now acceptable for publication, you may indicate that here to bypass the “Comments to the Author” section, enter your conflict of interest statement in the “Confidential to Editor” section, and submit your "Accept" recommendation.

Reviewer #1: (No Response)

Reviewer #2: (No Response)

2. Is the manuscript technically sound, and do the data support the conclusions?

Reviewer #1: Yes

Reviewer #2: Yes

3. Has the statistical analysis been performed appropriately and rigorously? 

Reviewer #1: Yes

Reviewer #2: Yes

4. Have the authors made all data underlying the findings in their manuscript fully available?

Reviewer #1: Yes

Reviewer #2: Yes

5. Is the manuscript presented in an intelligible fashion and written in standard English?

Reviewer #1: Yes

Reviewer #2: Yes

6. Review Comments to the Author

Reviewer #1: Comments to authors

General comments.

The manuscript has been considerably improved by the authors following the first round of comments. However, there are some issues remaining to be resolved. They are detailed below.

Note. The line numbers were not working in many cases. This made difficult to trace the corrections and has to be corrected in the next versions.

Introduction

Line 134. What have found to be the role of small ruminant in the epidemiology of FMD? Are they carrier transmitter, spill over hosts, primary host which can maintain the disease by themselves irrespective the presence of cattle or what? I don’t think the scope of the work can lead to these types of conclusions.

Material and methods

Line 184. A herd…..

Line 187. Instead of the software, provide reference for methodology of the sample size determination. Moreover, the website. http://www.promesa.co.nzl seems not working. check it.

Line 188. ‘expected between herd prevalence 30%” what does it mean and how is the 30% determined?

Line 190. Where did your get the intracluster correlation is 0.2?

Line 193. “Prevalence in a herd is 20%”. Give reference from where did you get this?

Generally, as which sample size formula is used is not mentioned, it is difficult to understand how sample size was determined.

Line 198 . remove” Because it was one village per sub-location and one herd per village, it is appropriate to use the terms interchangeably” as it is confusing which terms it refers, and it also adds no clarity.

Line 224-227. multistage sampling; sublocation, village, herd and then individual animals. Compare this with your description in line 181-183.

Line 308. (Pearson?) correlation coefficient of 0.3 threshold for collinearity is highly conservative. Are there any references for this?

Multicollinearity would have been more appropriate than simple collinearity.

Results

Line 334. Herd was defined better in M& M and not needed here.

Line 344. design effect for 2 may not be high for FMD as it id highly transmitted within a herd and could result high clustering effect.

Line 403. “None of the herds surveyed had animals that showed FMD signs” . this should be qualified as at the time of sampling.

Line 447. What are you doing in the above two tables? were you not assessing association of risk factors for SR FMD? When you say FMD seropositivity was higher in adult than young, don’t you mean age is a risk factor?! So the above analysis a univariable (crude) risk factor analysis and should come under risk factors analysis title. Actually, I don’t see its importance as far as you do the multivariable analysis as in table 5. After all the definitive test for risk factor is the multivariable test. This has been commented in the previous version.

Line 489. Table 5. This is the final model. Then why was the non-significant factors such as Brought in SR, SR breeding method, elevation included in the final model. In the methodology it was mentioned backward fitting (selection?) was used. If that is the case, in backward selection at each iteration the non-significant variables are removed until only significant factors and confounding factors remain in the final model.

Line 513. The interpretation for interaction is not clear. After the doing the interaction I would expect results such as: maleness and adultness interact positively and increased risk of FMD more than either of maleness or adultness alone.

Discussion

Line 534. Across the categories

Line 583-684. Any evidence about subclinical infection SR to conclude this?

Conclusion

Line 600. What is this role?

Line 600 -604 your result could not give any clue about potential transmission of FMD between cattle and sheep let alone sheep as reservoir of infection for cattle. Is it not possible SR are getting infection from cattle? It is known that cattle are more susceptible to FMD and are often the maintenance hosts for FMD viruses except SAT viruses where African buffalo are involved. Cattle tend to be carrier longer and only short period of carrier was reported for SR.

This comment has been given in previous version as well but not has not been corrected. So should be corrected or need convincing explanation.

Reviewer #2: The authors have done a good job in working on the previous reviewer’s comments. Unfortunately, it seems some of my specific comments from review round 1 were not sent forward to the authors and therefore some that have not already addressed are repeated here again below.

Although the manuscript has been shortened, I still think there is a requirement to further make the introduction and results more concise, which will make the study more pleasurable to read, as the results are good. I think there are options for tables to be combined and the number of columns reduced so that the important results do not get lost (see specific comments below).

Additionally, from what I can see there is no description as to what S2 file 1 represents and therefore I think it should be removed.

Specific comments

Abstract

Line 31-32 – No need to capitalise ‘non-structural proteins’. And remove the second full stop after ‘kit’.

Line 37 – Please provide the p number – not just say that it is was less than 0.05. Also, I think just the p value is enough, no need to include X2 value). However, I think there is no need to include this sentence (‘Sero-positivity was significantly……..p<0.05)’, as you have stated below that the sedentary production zone was negatively associated with seropositivity (line 40-41).

Introduction

Needs shortening further and moving a few things around to make more concise. A focus on Kenya and small ruminants is recommended.

Line 72 – I do not think there is a need for Table 1. Or if so it should be included in the supplementary files. I think a short sentence about FMD prevalence on other East African countries would be enough here.

Lines 66-70 – Suggest removing this short paragraph, and instead mentioning seroprevalence studies of FMD in small ruminants in other East African countries.

Lines 75-108 – suggest reducing this further to make the introduction more concise. I think the most important thing to mention is FMD in small ruminants. (serotypes in Kenya are already mentioned below so no need to mention here?)

Lines 109-119 – Suggest removing this paragraph.

Lines 124 – 127 – Suggest removing these two sentences (‘Vaccination programmes…….maternal immunity’).

Lines 135 – 137 – suggest removing this sentence (‘This is because…..borders’).

Methods

Lines 151-158 – I think that this paragraph can be removed?

Line 161 – Add a full stop after ‘country’.

Figure 1 – I’m not sure if it is just the version that is attached to this PDF, but please ensure Figure 1 is clear, as the version I can see is blurred.

Lines 183 – 184 – Suggest removing these sentences (‘The total sample……strata’). Also suggest using the term zone instead of strata going forward.

Line 184 – please do not capitalise the word ‘herd’.

Line 188-189 – please state why a 30% expected between herd prevalence was used, and in line 193, why an expected prevalence of FMD in the herd of 20% was used. What information was this based on?

Lines 214-216 – suggest removing these two first sentences as this information has already been described previously.

Line 218 – change to ‘these WERE used…’

Line 279 – 280 – Were two different ELISA tests performed or is the ‘3ABC blocking enzyme-linked immunosorbent assay’ the same as the ‘3 ABC- ELISA ID Screen®FMD NSP Competition kit’. I assume they are the same, but I think it is confusing here. Maybe it is best to say in line 275 that ‘Individual animal serum samples were analysed……’ and then remove the sentence in line 279-280 (‘At individual level……assay’).

Line 287 – I do not think S2 file should be included – it is not very self-explanatory, and all of the data are in the other supplementary files.

Line 295 – What does this refer to? ‘--“Bunny-Wunnies Freak Out”’? Please remove.

Results

Line 340 – Please do not capitalise ‘counties’

Lines 341-345 – Suggest moving this sentence and reasoning to the discussion.

Suggest removing Table 2, as the important information is displayed in the other tables. Move to supplementary data if necessary to keep it. I am not sure what the column ‘sum’ refers to in Table 2. Is this table referring to the herd sizes that animals were sampled from, or the animals sampled?

Lines 355-398 – This section needs to be made more concise – suggest reducing to one paragraph. If the Table is brought to the main article instead of supplementary, only a short summary of results is required.

Lines 403-405 – Please improve the English language of this sentence.

I’m not sure that confidence intervals are required for apparent sero-prevalence – only for true prevalence.

Table 3 – remove variable code. Suggest to remove the ‘negative’ column. Suggest adding p value column for each variable in Table 3 for chi squared test. Then the reader can clearly see the results without hunting through the text.

Table 4 – remove variable code. Suggest to remove the ‘negative’ column. Suggest adding p value column for each variable in Table 4 for chi-squared test. Then the reader can clearly see the results without hunting through the text.

Can Tables 3 and 4 be combined?

Lines 403-474 – Remove ꭓ2 values from the text and only include p values. Also this section needs to be made more concise. Including the p values for the chi-squared tests in Tables 3 and 4 would mean less description is required.

Line 479 – Do ‘all the potential risk factors’ mean all of those in Tables 3 and 4? Please be clear.

Table 5 – Combine with Table 6? There are too many unnecessary columns. Suggest removing the columns B, Se/S.E. and Z ratio.

Discussion

Line 534 – ‘categories’ is spelled incorrectly.

Line 541 – change ‘FMD outbreak peaks is…’ to ‘FMD outbreaks are associated…’.

Line 545 – Although foot and mouth disease is spread by contact, I would recommend changing this sentence as other methods of transmission are also common e.g. fomites/aerosol. It is more likely that as herd are closer together there are more chances for the virus to spread?

Line 581 – ‘transmission’ is spelled incorrectly.

Suggestion to include a discussion about the limitation of the study that testing was not done for circulating virus (e.g. virus isolation/PCR), and therefore it is unknown when these animals became infected. Indeed, further investigations (potentially using the same samples) could be done and also to identify the serotypes that the SR were infected with using serotype specific ELISAs.

Line 606 – Remove double word ‘cross’.

7. PLOS authors have the option to publish the peer review history of their article (what does this mean?). If published, this will include your full peer review and any attached files.

Reviewer #1: No

Reviewer #2: No

---

## [Author Response · Author response to Decision Letter 1]

5 Jul 2021

We are sincerely grateful to both Reviewers for the valuable comments and suggetions that have helped us improve the manuscript and sharpen our writing skills. Below are our responses to the comments by the reviewers.

Reviewer #1: Comments to authors

General comments.

The manuscript has been considerably improved by the authors following the first round of comments. However, there are some issues remaining to be resolved. They are detailed below.

Note. The line numbers were not working in many cases. This made difficult to trace the corrections and has to be corrected in the next versions.

Thank you for the compliment. We apologize for the hardship faced in tracing corrections. In this revision, reference is made to the line numbers in the revised manuscript with track changes for ease of traceability. However the line numbers in the manuscript with accepted changes may be quite different because deleted lines in the revised manuscript with track changes remain while deleted lines in the manuscript with accepted changes disappear.

Introduction

Line 134. What have found to be the role of small ruminant in the epidemiology of FMD? Are they carrier transmitter, spill over hosts, primary host which can maintain the disease by themselves irrespective the presence of cattle or what? I don’t think the scope of the work can lead to these types of conclusions.

Statement has been removed in L139-143.

Material and methods

Line 184. A herd…..

Done on line 190.

Line 187. Instead of the software, provide reference for methodology of the sample size determination. Moreover, the website. http://www.promesa.co.nzl seems not working. check it.

Reference for formula used provided on Line 193.

Line 188. ‘expected between herd prevalence 30%” what does it mean and how is the 30% determined?

Corrected and reference provided on Line 196.

Line 190. Where did your get the intracluster correlation is 0.2?

Corrected and reference provided on Line 197.

Line 193. “Prevalence in a herd is 20%”. Give reference from where did you get this?

Reference provided on Line 203.

Generally, as which sample size formula is used is not mentioned, it is difficult to understand how sample size was determined.

Provided in Line 200-205.

Line 198. remove” Because it was one village per sub-location and one herd per village, it is appropriate to use the terms interchangeably” as it is confusing which terms it refers, and it also adds no clarity.

Removed in Line 209-210. 

Line 224-227. multistage sampling; sublocation, village, herd and then individual animals. Compare this with your description in line 181-183.

Content of Line 224-227 has been removed in Line 236-238 to avoid confusion.

Line 308. (Pearson?) correlation coefficient of 0.3 threshold for collinearity is highly conservative. Are there any references for this?

Multicollinearity would have been more appropriate than simple collinearity.

It is acceptable to measure multicollinearity by the variance inflation factor (VIF) for continuous data or Spearman correlation for categorical data. Our data being mainly categorical could not utilize the VIF method. Reference [54] for correlation coefficient threshold is offered in Line 327. 

Results

Line 334. Herd was defined better in M& M and not needed here.

Removed in Line 353.

Line 344. design effect for 2 may not be high for FMD as it id highly transmitted within a herd and could result high clustering effect.

Corrected to ‘sufficient’ on Line 611.

Line 403. “None of the herds surveyed had animals that showed FMD signs”. This should be qualified as at the time of sampling.

Done in Line 447-448.

Line 447. What are you doing in the above two tables? Were you not assessing association of risk factors for SR FMD? When you say FMD sero-positivity was higher in adult than young, don’t you mean age is a risk factor?! So the above analysis a univariable (crude) risk factor analysis and should come under risk factors analysis title. Actually, I don’t see its importance as far as you do the multivariable analysis as in table 5. After all the definitive test for risk factor is the multivariable test. This has been commented in the previous version.

The study is not just about association. It is important to know what the sero-positivities were across the categories with or without association and then find the association with the potential risk factors as the title indicates. It is acceptable to compare proportions without necessarily establishing association and strength of association with independent variables but in our study we have done both according to our objective.

Line 489. Table 5. This is the final model. Then why was the non-significant factors such as Brought in SR, SR breeding method, elevation included in the final model. In the methodology it was mentioned backward fitting (selection?) was used. If that is the case, in backward selection at each iteration the non-significant variables are removed until only significant factors and confounding factors remain in the final model.

The most parsimonious model together with interactions/contrasts has now been presented in Table 5.

Line 513. The interpretation for interaction is not clear. After the doing the interaction I would expect results such as: maleness and adultness interact positively and increased risk of FMD more than either of maleness or adultness alone.

This has been clarified in Line 585 to 591.

Discussion

Line 534. Across the categories.

Corrected in Line 676.

Line 583-684. Any evidence about subclinical infection SR to conclude this?

Do you mean Line 583-584?

Sentence has been removed in Line 683-687. 

Conclusion

Line 600. What is this role?

Line 600 -604 your result could not give any clue about potential transmission of FMD between cattle and sheep let alone sheep as reservoir of infection for cattle. Is it not possible SR are getting infection from cattle? It is known that cattle are more susceptible to FMD and are often the maintenance hosts for FMD viruses except SAT viruses where African buffalo are involved. Cattle tend to be carrier longer and only short period of carrier was reported for SR.

This comment has been given in previous version as well but not has not been corrected. So should be corrected or need convincing explanation.

The content of these lines has been recast in the revised manuscript and unjustified assertions removed.

Reviewer #2: The authors have done a good job in working on the previous reviewer’s comments. 

Thank you for the compliment.

Unfortunately, it seems some of my specific comments from review round 1 were not sent forward to the authors and therefore some that have not already addressed are repeated here again below.

Although the manuscript has been shortened, I still think there is a requirement to further make the introduction and results more concise, which will make the study more pleasurable to read, as the results are good. I think there are options for tables to be combined and the number of columns reduced so that the important results do not get lost (see specific comments below).

Additionally, from what I can see there is no description as to what S2 file 1 represents and therefore I think it should be removed.

Specific comments

Abstract

Line 31-32 – No need to capitalise ‘non-structural proteins’. And remove the second full stop after ‘kit’.

Done in Line 32-33.

Line 37 – Please provide the p number – not just say that it is was less than 0.05. Also, I think just the p value is enough, no need to include X2 value). However, I think there is no need to include this sentence (‘Sero-positivity was significantly……..p<0.05)’, as you have stated below that the sedentary production zone was negatively associated with seropositivity (line 40-41).

Sentence has been removed in Line 36-37.

Introduction

Needs shortening further and moving a few things around to make more concise. A focus on Kenya and small ruminants is recommended.

Line 72 – I do not think there is a need for Table 1. Or if so it should be included in the supplementary files. I think a short sentence about FMD prevalence on other East African countries would be enough here.

Table 1 taken out to be supplementary and sentence on prevalence of FMD in East African countries inserted in Line 71-75.

Lines 66-70 – Suggest removing this short paragraph, and instead mentioning seroprevalence studies of FMD in small ruminants in other East African countries.

Removed in Line 66-68 and summary sentence on sero-prevalence studies in East African countries inserted in Line 71-75.

Lines 75-108 – suggest reducing this further to make the introduction more concise. I think the most important thing to mention is FMD in small ruminants. (serotypes in Kenya are already mentioned below so no need to mention here?)

Removed in Line 89-92.

Lines 109-119 – Suggest removing this paragraph.

Removed in Line 115-126.

Lines 124 – 127 – Suggest removing these two sentences (‘Vaccination programmes…….maternal immunity’).

Removed in Line 130-133.

Lines 135 – 137 – suggest removing this sentence (‘This is because…..borders’).

Removed in Line 141-143.

Methods

Lines 151-158 – I think that this paragraph can be removed?

Removed in Line 157-164.

Line 161 – Add a full stop after ‘country’.

Done in Line 167.

Figure 1 – I’m not sure if it is just the version that is attached to this PDF, but please ensure Figure 1 is clear, as the version I can see is blurred.

I think it is the pdf version that is blurred.

Lines 183 – 184 – Suggest removing these sentences (‘The total sample……strata’). Also suggest using the term zone instead of strata going forward.

Done in Line 189-190. Zone applied instead of strata going forward.

Line 184 – please do not capitalise the word ‘herd’.

Done in Line 190.

Line 188-189 – please state why a 30% expected between herd prevalence was used, and in line 193, why an expected prevalence of FMD in the herd of 20% was used. What information was this based on?

Given in Line 196 and 203.

Lines 214-216 – suggest removing these two first sentences as this information has already been described previously.

Removed in Line 209-210.

Line 218 – change to ‘these WERE used…’

The ‘WAS’ is referring to the sampling frame which is one and not the sub-locations and clarified accordingly on Line 229.

Line 279 – 280 – Were two different ELISA tests performed or is the ‘3ABC blocking enzyme-linked immunosorbent assay’ the same as the ‘3 ABC- ELISA ID Screen®FMD NSP Competition kit’. I assume they are the same, but I think it is confusing here. Maybe it is best to say in line 275 that ‘Individual animal serum samples were analysed……’ and then remove the sentence in line 279-280 (‘At individual level……assay’).

Done in Line 287/88 and Line 291-293.

Line 287 – I do not think S2 file should be included – it is not very self-explanatory, and all of the data are in the other supplementary files.

File has been removed in Line 300.

Line 295 – What does this refer to? ‘--“Bunny-Wunnies Freak Out”’? Please remove.

Removed in Line 309.

Results

Line 340 – Please do not capitalise ‘counties’

Corrected in Line 359/360.

Lines 341-345 – Suggest moving this sentence and reasoning to the discussion.

Moved to line 607-612.

Suggest removing Table 2, as the important information is displayed in the other tables. Move to supplementary data if necessary to keep it. I am not sure what the column ‘sum’ refers to in Table 2. Is this table referring to the herd sizes that animals were sampled from, or the animals sampled?

The term ‘sum’ refers to number of sheep and goats in the herds sampled in the study area as indicated in the title but the term has been revised in the table and the title has been made clearer. The data in this table is not in any other table and therefore cannot be removed but is now placed as supplementary table 2.

Lines 355-398 – This section needs to be made more concise – suggest reducing to one paragraph. If theTable is brought to the main article instead of supplementary, only a short summary of results is required.

There are two Tables in this text which have now been brought to the main article (Table 1 and 2) and summaries given (Line 376-379 and Line 391-399). The tables cannot be combined because it would result in too large a table.

Lines 403-405 – Please improve the English language of this sentence.

Done in Line 447-449.

I’m not sure that confidence intervals are required for apparent sero-prevalence – only for true prevalence.

Because of the near perfection of the tests (Se=100%); Sp=99%). Calculation of the CIs of true prevalence p±1.96√(pq/(nJ^2 )) is equivalent to that of simple proportion p±1.96√pq/n as J (Se+Sp-1) nears 1. Thus CIs of true prevalence are similar to those of apparent prevalence and have been applied to the section in line 450-465.

Table 3 – Remove variable code. Suggest to remove the ‘negative’ column. Suggest adding p value column for each variable in Table 3 for chi squared test. Then the reader can clearly see the results without hunting through the text.

Table 4 – remove variable code. Suggest to remove the ‘negative’ column. Suggest adding p value column for each variable in Table 4 for chi-squared test. Then the reader can clearly see the results without hunting through the text.

In Table 3 and 4. Done for overall chi-square for each variable but for those with more than two categories, pairwise chi-square results have been given in the text as they cannot be included in the tables.

Can Tables 3 and 4 be combined?

No because combining them will create too large a table. 

Lines 403-474 – Remove ?2 values from the text and only include p values. Also this section needs to be made more concise. Including the p values for the chi-squared tests in Tables 3 and 4 would mean less description is required. Done. However, for pairwise comparisons, chi-square remain in the text as they cannot be expressed in the Tables.

Line 479 – Do ‘all the potential risk factors’ mean all of those in Tables 3 and 4? Please be clear.

Clarified in Line 553.

Table 5 – Combine with Table 6? There are too many unnecessary columns. Suggest removing the columns B, Se/S.E. and Z ratio.

Table 5 combined with Table 6 and corrections made as suggested. 

Discussion

Line 534 – ‘categories’ is spelled incorrectly.

Corrected in line 676.

Line 541 – change ‘FMD outbreak peaks is…’ to ‘FMD outbreaks are associated…’.

Corrected on Line 631.

Line 545 – Although foot and mouth disease is spread by contact, I would recommend changing this sentence as other methods of transmission are also common e.g. fomites/aerosol. It is more likely that as herd are closer together there are more chances for the virus to spread?

Corrected on Line 635-636.

Line 581 – ‘transmission’ is spelled incorrectly.

Corrected in Line 683.

Suggestion to include a discussion about the limitation of the study that testing was not done for circulating virus (e.g. virus isolation/PCR), and therefore it is unknown when these animals became infected. Indeed, further investigations (potentially using the same samples) could be done and also to identify the serotypes that the SR were infected with using serotype specific ELISAs.

Included in line 702-712.

Line 606 – Remove double word ‘cross’.

Removed in line 728.

7. PLOS authors have the option to publish the peer review history of their article (what does this mean?). If published, this will include your full peer review and any attached files.

Do you want your identity to be public for this peer review? For information about this choice, including consent withdrawal, please see our Privacy Policy.

Reviewer #1: No

Reviewer #2: No

---

## [Editor Report · Decision Letter 2]

8 Jul 2021

Epidemiological study on foot-and-mouth disease in small ruminants: sero-prevalence and risk factor assessment in Kenya

PONE-D-20-15166R2

Dear Dr. Kairu-Wanyoike,

We’re pleased to inform you that your manuscript has been judged scientifically suitable for publication and will be formally accepted for publication once it meets all outstanding technical requirements.

Kind regards,

Jagadeesh Bayry, DVM, PhD, HDR

Academic Editor

PLOS ONE
---

## [Editor Report · Acceptance letter]

22 Jul 2021

PONE-D-20-15166R2 

Epidemiological study on foot-and-mouth disease in small ruminants: sero-prevalence and risk factor assessment in Kenya 

Dear Dr. Kairu-Wanyoike:

I'm pleased to inform you that your manuscript has been deemed suitable for publication in PLOS ONE. Congratulations! Your manuscript is now with our production department. 

Kind regards, 

on behalf of

Dr. Jagadeesh Bayry 

Academic Editor

PLOS ONE